# Developmental programming modulates olfactory behavior in *C. elegans* via endogenous RNAi pathways

Jennie R Sims[1†], Maria C Ow[1†], Mailyn A Nishiguchi[1], Kyuhyung Kim[2], Piali Sengupta[3], Sarah E Hall[1]*

[1]Department of Biology, Syracuse University, Syracuse, United States; [2]Department of Brain and Cognitive Sciences, Daegu Gyeongbuk Institute of Science and Technology, Daegu, Republic of Korea; [3]National Center for Behavioral Genomics, Department of Biology, Brandeis University, Waltham, United States

**Abstract** Environmental stress during early development can impact adult phenotypes via programmed changes in gene expression. *C. elegans* larvae respond to environmental stress by entering the stress-resistant dauer diapause pathway and resume development once conditions improve (postdauers). Here we show that the *osm-9* TRPV channel gene is a target of developmental programming and is down-regulated specifically in the ADL chemosensory neurons of postdauer adults, resulting in a corresponding altered olfactory behavior that is mediated by ADL in an OSM-9-dependent manner. We identify a *cis*-acting motif bound by the DAF-3 SMAD and ZFP-1 (AF10) proteins that is necessary for the differential regulation of *osm-9,* and demonstrate that both chromatin remodeling and endo-siRNA pathways are major contributors to the transcriptional silencing of the *osm-9* locus. This work describes an elegant mechanism by which developmental experience influences adult phenotypes by establishing and maintaining transcriptional changes via RNAi and chromatin remodeling pathways.

*For correspondence: shall@syr.edu

[†]These authors contributed equally to this work

**Competing interests:** The authors declare that no competing interests exist.

## Introduction

Increasing evidence suggests that exposure to stressful environmental conditions during a critical period in the early development of an organism can result in altered gene expression patterns that are maintained into adulthood. In mammals, this 'cellular memory' of early life stress has been shown to correlate with behavioral phenotypes that persist into adulthood (*Weaver et al., 2004b*; *McGowan et al., 2009*). For example, transcription levels of the *glucocorticoid receptor* (GR) gene in the mouse hippocampus are programmed by the maternal care behaviors of the mother towards her pups during the first postnatal week (*Fish et al., 2004*; *Weaver et al., 2004a*; *2007*; *Szyf, 2007*; *2008*). High maternal care behaviors (such as frequent licking and grooming) lead to high GR expression and low stress responses in the pups as adults, whereas low maternal care results in increased DNA methylation, transcriptional silencing of the GR gene, and high stress responses. Intriguingly, this mechanism of gene regulation appears to be conserved in humans, as childhood adversity correlates with increased transcriptional silencing of the GR gene and mental disorders in adults (*McGowan et al., 2008*; *2009*; *Labonté et al., 2012*; *Labonte et al., 2012*). However, the molecular mechanisms establishing and maintaining the tissue-specific changes in gene expression in neurons due to environmental inputs during early development remain to be elucidated.

We have shown that environmental stress early in development can also modulate gene expression and behavior in *Caenorhabditis elegans,* thereby establishing nematodes as a model system for investigating the mechanisms underlying developmental programming of gene expression

**eLife digest** Increasing evidence suggests that experiencing stressful environments early on in life can have profound effects on the health and behavior of adults. For example, stressful conditions in the womb have been linked to adult depression and metabolic disorders. These effects are thought to be the result of changes in the way that genes in specific tissues are regulated in the individuals that have experienced the stress. However, it is not clear how a particular stress can cause long-term changes in gene activity in specific tissues.

A microscopic worm called *Caenorhabditis elegans* is often used as a simple animal model to study how animals develop and behave. Previous studies have shown that adult worms that experienced stress early in life show differences in behavior and gene activity compared to genetically identical worms that did not experience the stress. Here, Sims, Ow et al. asked what signals are required for these changes to happen.

The experiments show that a gene called *osm-9* – which plays a role in the nervous system – is less active in sensory nerve cells in worms that experienced stress early on in life. This loss of activity resulted in the worms being unable to respond to a particular odor. Two proteins called DAF-3 and ZFP-1 are able to bind to a section of DNA in the *osm-9* gene to decrease its activity in response to stress. These proteins are similar to human proteins that are important for development and are associated with some types of leukemia. Further experiments show that small molecules of ribonucleic acid in the "RNA interference" pathway also help to decrease the activity of *osm-9* after stress.

Together, Sims, Ow et al.'s findings suggest that environmental conditions in early life regulate the *osm-9* gene through the coordinated effort of DAF-3, ZFP-1 and the RNA interference pathway. The next steps are to investigate how these molecules are able to target *osm-9* and to identify other proteins that regulate gene activity in response to stress in early life.

(*Hall et al., 2010*; *2013*). The *C. elegans* life cycle is regulated by environmental cues during a critical period in the first larval stage. If conditions are favorable (low temperature, high food availability, low pheromone concentrations), worms continue to develop through four larval stages to become reproductive adults (control adults, CON). However, if conditions are unfavorable, worms enter the alternative stress-induced dauer diapause stage (*Golden and Riddle, 1982*; *1984*). The dauer formation (*daf*) decision is regulated by the differential expression of conserved insulin and TGF-β pathways in sensory neurons in response to environmental conditions (reviewed in *Fielenbach and Antebi, 2008*). Once conditions improve, dauer animals re-enter the reproductive cycle and continue development to become reproductive adults (postdauer adults, PD). We have shown previously that CON and PD adults retain a cellular memory of their developmental history through changes in gene expression, genome-wide chromatin state, and life history traits (*Hall et al., 2010*). In addition, these altered adult phenotypes are in part dependent on endogenous small interfering RNA pathways (endo-siRNAs) (*Hall et al., 2013*).

In recent years, our knowledge of *C. elegans* endo-siRNA pathways and their gene targets has increased substantially. Endo-siRNAs are ~20 to 30 nt in length, antisense to coding transcripts, and are categorized into two major groups based on their biogenesis pathways (*Gent et al., 2010*; *Vasale et al., 2010*). Primary endo-siRNAs are 26 nt long, have a characteristic 5' monophosphate, and are derived from Dicer processing of double-stranded RNA template (26G-siRNAs) (*Vasale et al., 2010*). In somatic tissue, biogenesis of ERGO-1 class 26G-siRNAs is dependent upon the enhanced RNAi (ERI) complex, which consists of the core proteins RNA-dependent RNA polymerase (RdRP) RRF-3, Dicer-related helicase DRH-3, and Tudor domain-containing protein ERI-5 that associate with Dicer DCR-1 (*Duchaine et al., 2006*; *Pavelec et al., 2009*; *Gent et al., 2010*; *Thivierge et al., 2012*). Through an unknown mechanism, 26G-siRNAs stimulate the production of secondary endo-siRNAs, which are 22 nt in length, have a 5' triphosphate, and are produced through reverse transcription of mRNA templates by RdRPs (22G-siRNAs) (*Smardon et al., 2000*; *Simmer et al., 2002*; *Maine et al., 2005*; *Vought et al., 2005*; *Aoki et al., 2007*; *Pak et al., 2007*; *She et al., 2009*; *Vasale et al., 2010*; *Pak et al., 2012*). Downstream effector functions of 22G-

siRNAs, such as mRNA degradation or chromatin remodeling, are mediated by their associated _worm-specific Argonaute_ protein (WAGO) in the cytoplasm or nucleus (_Yigit et al., 2006_). Additionally, _Mutator_ proteins have been shown to be essential for the biogenesis and siRNA amplification of AGO ERGO-1 class 26G- and WAGO class 22G-siRNAs (_Zhang et al., 2011_; _Phillips et al., 2012_; _2014_). Although components of the _Mutator_ complex are expressed throughout the worm, their functions and cellular localization are specific to the germline or soma (_Phillips et al., 2012_; _2014_). Despite our understanding of endo-siRNA biogenesis, how siRNAs target and regulate endogenous genes, particularly in response to environmental cues, is poorly understood.

In this study, we show that expression of the OSM-9 TRPV channel is differentially regulated based on the developmental trajectory of _C. elegans_ and describe the molecular mechanisms that regulate expression of this gene as a function of the animal's developmental history. The _osm-9_ gene is down-regulated specifically in the ADL chemosensory neurons of PD adults, resulting in a corresponding altered olfactory behavior that is mediated by ADL in an OSM-9-dependent manner. We identify a _cis_-acting motif bound by the DAF-3 SMAD and ZFP-1 (AF10) proteins that is necessary for the down-regulation of _osm-9_ in PD adults and demonstrate that both chromatin remodeling and endo-siRNA pathways are major contributors to _osm-9_ regulation. Our results suggest a mechanism of _osm-9_ regulation whereby the dauer developmental decision, triggered by environmental stress, results in transcriptional silencing of the _osm-9_ locus mediated by the TGF-β, chromatin remodeling, and endogenous RNAi pathways. This work describes an elegant mechanism of how tissue-specific changes in gene expression triggered by transient environmental cues can be maintained via developmental programming.

## Results

### Expression of the _osm-9_ TRPV gene is regulated by developmental history

Gene expression analyses identified the _osm-9_ TRPV channel gene as a candidate gene whose expression is regulated as a consequence of passage through the dauer stage (_Hall et al., 2010_). _osm-9_ is expressed in a subset of head and tail neurons and is required for chemosensory, osmosensory, and mechanosensory behaviors in adult animals (_Colbert et al., 1997_; _de Bono et al., 2002_; _Jansen et al., 2002_; _Tobin et al., 2002_; _White et al., 2007_; _O'Halloran et al., 2009_; _Sassa et al., 2013_; _Wang et al., 2015a_). To characterize the differential gene expression of _osm-9_ between CON and PD adult animals in further detail, we examined the expression pattern of an integrated _gfp_ reporter gene driven by 375 bp of _osm-9_ upstream regulatory sequences and 18 bp of the first exon (_osm-9_p::_gfp_) (_Figures 1A_, _2A_). We examined _gfp_ expression in CON and PD adults that spent 24 hr in dauer induced by crowding (see Materials and Methods) (_Hall et al., 2010_). We observed GFP in the majority of AWA and ADL sensory neurons in CON animals (_Figures 1A,B_; _Figure 1—figure supplement 1A_). However, while expression in AWA neurons was unaffected, GFP expression was significantly down-regulated in ADL neurons of PD adults (_Figures 1A,B_; _Figure 1—figure supplement 1A_). These results suggest that _osm-9_ is regulated at the transcriptional level as a consequence of passage through the dauer stage.

To confirm that the differential expression of _osm-9_p::_gfp_ was reflective of the endogenous _osm-9_ gene, we quantified _osm-9_ mRNA molecules in CON and PD ADL neurons using single molecule florescent in situ hybridization (smFISH) (_Raj et al., 2008_; _Ji and van Oudenaarden, 2012_). We used a transgenic strain expressing GFP in ADL neurons under a promoter whose expression is unaffected upon dauer passage (_sre-1_p::_gfp_) to identify the correct neuron and to quantify the _gfp_ mRNA molecules present as a control (_Figure 1—figure supplement 3_). For both the _osm-9_ and _gfp_ smFISH probes, we observed that the fluorescent foci within ADL exhibited unequal sizes and intensities; thus, we were not confident that we could resolve the number of individual mRNA molecules simply by counting the number of smFISH loci. Instead, we quantified the fluorescence intensities of ADL neurons as a measure of the _osm-9_ and _gfp_ mRNA molecules present. We found that ADL neurons in CON adults exhibited significantly greater mean and maximum fluorescence intensities for the _osm-9_ mRNA probes compared to PD adults (_Figure 1C_ and _Figure 1—figure supplement 2_). In contrast, fluorescence intensities for the _gfp_ mRNA probes exhibited overall similar mean and maximum fluorescence levels for PD compared to CON adults (_Figure 1—figure supplement 3_). We

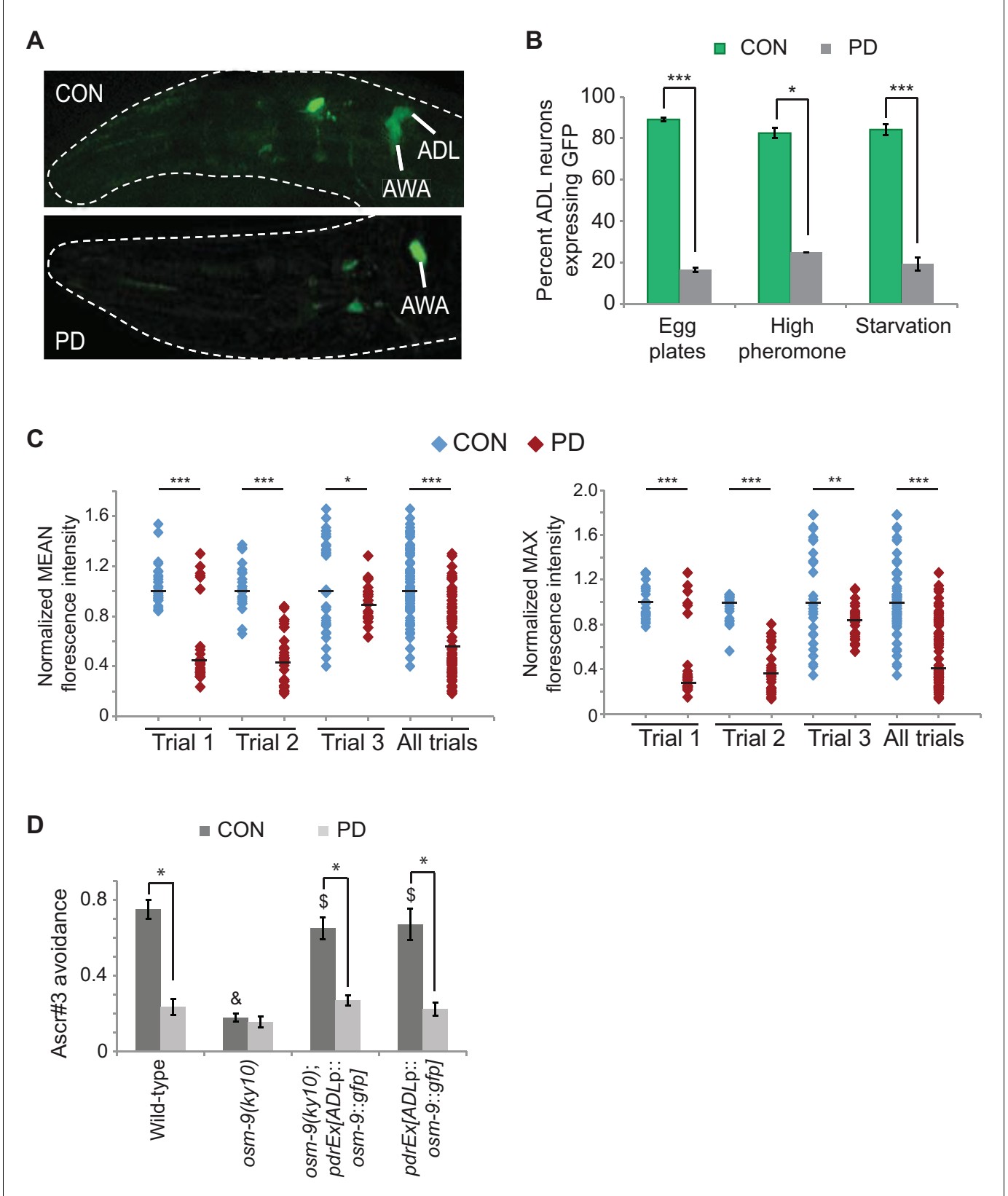

**Figure 1.** The *osm-9* TRPV channel gene is regulated by developmental history. (**A**) An *osm-9*p::*gfp* transgene is expressed in ADL and AWA neurons in CON adults, but is down-regulated in ADL neurons in PD adults. (**B**) *osm-9*p::*gfp* expression in ADL neurons in CON and PD adults that entered dauer

*Figure 1 continued*

due to crowding (egg plates), exposure to crude pheromone with low population density, or starvation. N ≥ 2 trials; n ≥ 40 animals (*Figure 1—source data 1*). **p<0.005, ***p<1 x 10$^{-9}$, Student's t-test. GFP in AWA is unaffected (*Figure 1—figure supplement 1*). (C) Quantification of *osm-9* mRNA in ADL neurons of CON and PD using smFISH. The graphs represent individual mean and maximum fluorescence measurements for N = 3 biologically independent trials; n ≥ 87 neurons (*Figure 1—source data 2*). Medians are indicated. ***p<1 x 10$^{-9}$, **p = 0.006 *p = 0.056, Student's t-test. See *Figure 1—figure supplement 2* and *Figure 1—figure supplement 3*. (D) Ascr#3 avoidance behavior of CON and PD adults normalized to M13 buffer. N ≥ 3 trials; n ≥ 60 animals (*Figure 1—source data 3*). Data for ADL-expressed *osm-9* rescue strains are averages of two extrachromosomal lines. * indicates CON significantly different from PD; & CON or PD significantly different from wild-type; $ CON or PD significantly different from *osm-9(ky10)*; One-way ANOVA with LSD posthoc correction, p<0.05. All error bars represent S.E.M. OSM-9 mediated behaviors modulated by ASH and AWA neurons are unaffected (*Figure 1—figure supplement 1* and *Figure 3—figure supplement 2*).

The following source data and figure supplements are available for figure 1:

**Source data 1.** Spreadsheet of percentages of animals expressing *osm-9*p::*gfp* in ADL neurons of wild-type strains that experienced overcrowding, high pheromone, or starvation conditions.
**Source data 2.** Spreadsheet containing mean and maximum florescence intensities of *osm-9* smFISH probes in wild-type ADL neurons.
**Source data 3.** Spreadsheet containing ascr#3 avoidance values for CON and PD in wild-type, *osm-9(ky10)*, and ADL-specific, *osm-9* rescue strains.
**Figure supplement 1.** OSM-9 expression in AWA was unaffected by developmental programming.
**Figure supplement 1—source data 1.** Spreadsheet containing percentage of animals expressing *osm-9*p::*gfp* in AWA neurons.
**Figure supplement 1—source data 2.** Spreadsheet containing chemotaxis indices in response to diacetyl for wild-type and *osm-9(ky10)* strains.
**Figure supplement 2.** *osm-9* mRNA levels are down-regulated in PD ADL neurons.
**Figure supplement 2—source data 1.** Spreadsheet containing mean and maximum florescence intensities for *osm-9* smFISH probes in ADL neurons and worm backgrounds.
**Figure supplement 3.** *gfp* mRNA levels are unaltered between CON and PD ADL neurons.
**Figure supplement 3—source data 1.** Spreadsheet containing normalized and raw mean and maximum florescence intensities for *gfp* smFISH probes in ADL neurons and worm backgrounds.

conclude that the differential regulation of the *osm-9*p::*gfp* transgene is reflective of the regulation of the endogenous *osm-9* gene.

Next, we asked whether passage through the dauer stage was necessary for the observed changes in gene expression, or if exposure to high concentrations of pheromone without dauer entry was sufficient as has been reported for the regulation of chemoreceptor genes in a subset of neurons in *C. elegans* (*Peckol et al., 2001*; *Nolan et al., 2002*). To address this question, we allowed wild-type embryos to hatch on plates containing crude pheromone as previously described (*Neal et al., 2013*). Animals that bypassed dauer entry and animals that entered the dauer stage were recovered separately and examined for *osm-9*p::*gfp* expression in adults. We found that animals that bypassed dauer entry when exposed to high concentrations of dauer pheromone continued to express GFP in ADL neurons, similar to CON animals in our egg plate preparation (*Figure 1B*). However, animals that passed through the dauer stage exhibited the expected decrease in *osm-9*p::*gfp* expression in ADL (*Figure 1B*). Similarly, animals that were transiently starved in early development continued to express GFP in ADL if they bypassed the dauer stage, but exhibited decreased GFP expression following passage through dauer (*Figure 1B*). These results indicate that the down-regulation of *osm-9*p::*gfp* in adults is dependent on passage through the dauer stage regardless of the environmental trigger inducing dauer entry.

## ADL-mediated behavior is altered as a consequence of passage through the dauer stage

To examine the phenotypic consequences of the differential regulation of *osm-9*, we tested whether an ADL-mediated and OSM-9 dependent behavior is affected upon passage through the dauer stage. Wild-type hermaphrodites avoid high concentrations of the dauer pheromone component ascr#3; this avoidance behavior requires the OSM-9 and OCR-2 TRPV channels in the ADL neurons (*Jang et al., 2012*). We hypothesized that if the endogenous *osm-9* gene were down-regulated in ADL neurons of PD animals, these animals would fail to avoid ascr#3 similar to *osm-9* mutants. To test this hypothesis, we examined the ability of wild-type and *osm-9* mutant CON and PD adults to avoid 100 nM ascr#3 using drop-test assays (*Hilliard et al., 2002*; *Jang et al., 2012*). We observed that wild-type PD adults exhibited a significantly reduced avoidance response to ascr#3 compared to CON adults (*Figure 1D*). However, these animals retained the ability to avoid glycerol, an OSM-9-dependent behavior mediated by the ASH neurons, and were attracted to the odorant diacetyl, an OSM-9-dependent behavior mediated by the AWA neurons (*Figure 1—figure supplement 1B*; *Figure 3—figure supplement 1C*) (*Sengupta et al., 1996*; *Colbert et al., 1997*). These observations are consistent with the observed down-regulation of *osm-9* expression in the ADL neurons.

To verify that the decreased ascr#3 avoidance phenotype in wild-type PD adults was due to decreased expression of *osm-9* in ADL neurons, we restored *osm-9* expression specifically in ADL neurons of *osm-9* mutants by driving the cDNA under *sre-1* upstream regulatory sequences. Since *sre-1* expression is not detectably altered upon passage through the dauer stage (*Figure 1—figure supplement 2A* and *Figure 1—figure supplement 3*), we predicted that both CON and PD adults expressing *sre-1*p::*osm-9* cDNA::*gfp* transgene would continue to avoid ascr#3. However, while the transgene rescued the *osm-9* mutant phenotype in CON animals indicating that the fusion protein is functional, avoidance behavior was again decreased in PD animals (*Figure 1D*). We also attempted to rescue the decrease in ascr#3 avoidance in wild-type PD adults by expressing the same *sre-1*p::*osm-9* cDNA::*gfp* transgene in ADL neurons. Again, we observed ascr#3 avoidance behavior similar to wild-type animals without the transgene (*Figure 1D*). Although our results thus far have indicated that *osm-9* is regulated at the transcriptional level in response to developmental history, these results suggest that silencing mechanisms might also target the *osm-9* coding sequences or mRNA, or alternatively, additional genes contributing to the ascr#3 behavior.

## *osm-9* down-regulation requires a *cis*-acting promoter motif

Since transcriptional down-regulation contributes to the decreased *osm-9*p::*gfp* expression in ADL in PD animals, we analyzed *osm-9* upstream regulatory sequences present in the *osm-9*p::*gfp* transgene to identify regulatory motifs contributing to the altered expression. Using bioinformatics-based analyses of *osm-9* upstream regulatory sequences, we identified a 29 bp motif (*Figure 2A*) that is present in the upstream regulatory sequences of 977 (4.9%) genes in the genome. This conserved sequence was slightly enriched (121 genes, 5.6%; Z-test, p=0.056) in the regulatory regions of genes that were differentially expressed between CON and PD adults (*Hall et al., 2010*). We also identified an additional motif, a predicted DAF-3 SMAD binding site, located 22 bp upstream of the conserved sequence (*Figure 2A*) (*Thatcher et al., 1999*). We will henceforth refer to this combined regulatory sequence as the PD motif. To test the functional relevance of this motif, we generated a series of *osm-9*p::*gfp* transcriptional fusion constructs lacking portions of the *osm-9* regulatory sequences and examined their expression in ADL neurons of CON and PD adults (*Figure 2A,C*). Deletion of the PD motif in the context of the 375 bp *osm-9* regulatory sequences (Deletion 1) resulted in continued expression of *gfp* in ADL in PD adults without affecting expression in CON animals, indicating that this *cis*-regulatory motif is required for the down-regulation of *osm-9* in PD ADL neurons (*Figure 2A, C*). Deletion of sequences that did not include the PD motif (Deletion 2) resulted in *gfp* expression changes in PD ADL neurons similar to that observed with the full length *osm-9*p::*gfp* transgene (*Figure 2A,C*). Deletion 3, which contained the 18 bp of *osm-9* coding sequence in addition to the 375 bp of upstream regulatory sequence, also exhibited down-regulation of *gfp* expression in ADL neurons of PD adults (*Figure 2A,C*). These results indicate that the *cis*-acting PD motif is necessary for the transcriptional regulation of *osm-9* due to developmental programming.

To further characterize the PD motif sequence necessary for differential regulation of *osm-9*, we sequentially replaced a subset of bases within the PD motif with guanines and examined *gfp*

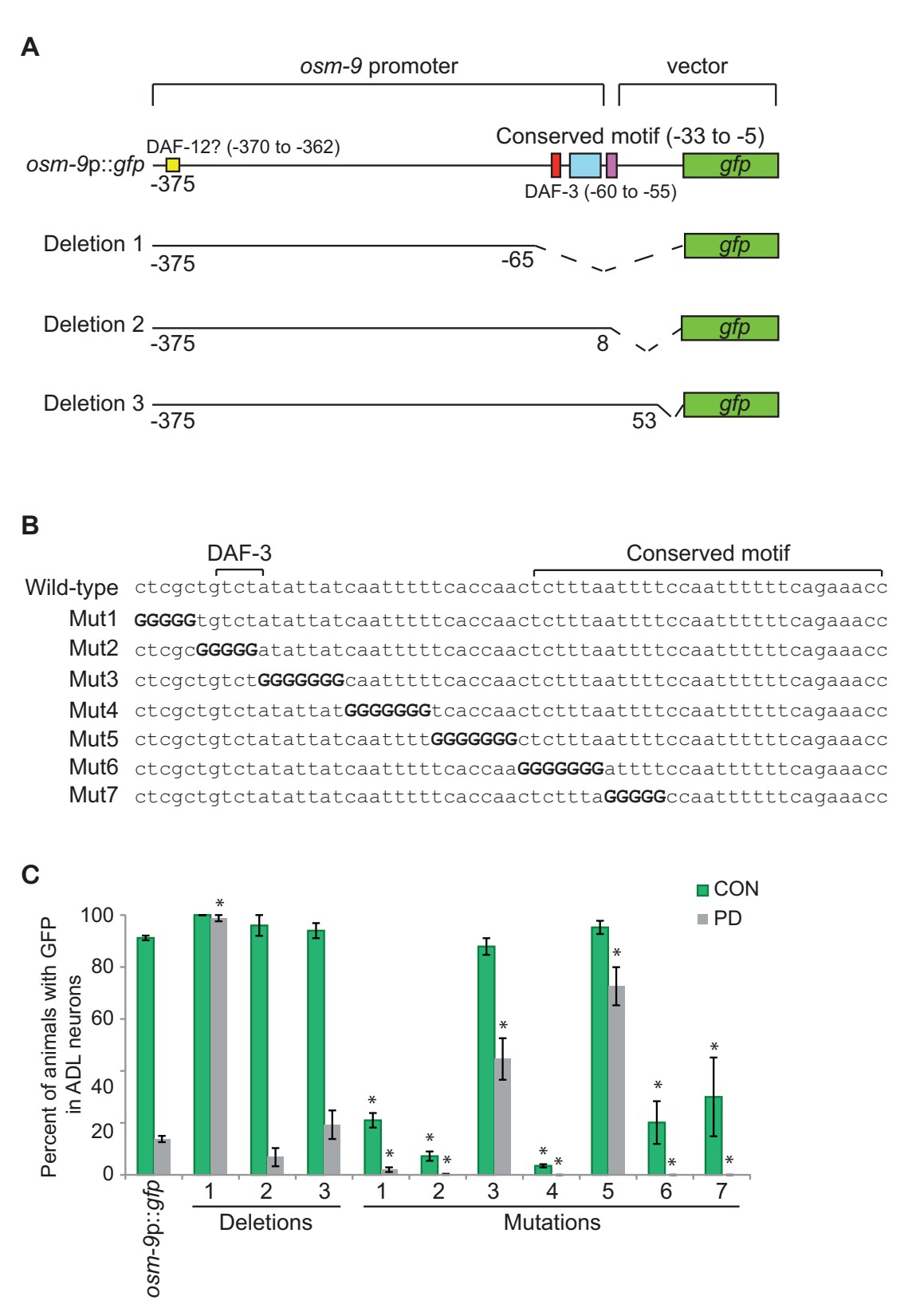

**Figure 2.** The PD motif is necessary for the differential regulation of *osm-9*. (**A**) Schematic of *osm-9* promoter driving *gfp* expression. Dotted lines indicate deleted sequences. Purple, blue, red, and yellow boxes represent 18 bp *osm-9* coding sequence, the conserved motif, DAF-3 binding site, and

*Figure 2 continued*

potential DAF-12 binding site, respectively. (**B**) The PD motif DNA sequences of wild-type and mutated versions. Bold 'G' indicates where wild-type sequence was mutated to guanine. (**C**) GFP expression in CON and PD ADL neurons of strains carrying deletion and mutated versions of *osm-9*p::*gfp*. Data for each deletion and mutation represents the average of two extrachromosomal lines. * indicates significantly different from wild-type CON or PD; One-way ANOVA with LSD posthoc correction, p<0.05. All error bars represent S.E.M (*Figure 2—source data 1*).
The following source data is available for figure 2:

**Source data 1.** Spreadsheet of percentages of animals expressing *gfp* in ADL neurons in strains carrying deletion and mutation versions of *osm-9*p::*gfp*.

expression in ADL neurons of CON and PD adults (*Figure 2B,C*). Mutated constructs 3 and 5 exhibited significantly increased GFP in ADL neurons of PD adults compared to wild-type (*Figure 2B,C*), indicating that these sequences were required for complete down-regulation of *osm-9* in PD adults. However, we found that mutated constructs 1, 2, 4, 6, and 7 resulted in a significant decrease in *osm-9*p::*gfp* expression in both CON and PD ADL neurons (*Figure 2B,C*), indicating that these sequences were required for the positive regulation of *osm-9* expression. Together these results suggest a model that, in the absence of the PD motif, the default state of *osm-9* expression in ADL neurons is transcriptional activation, and that the PD motif contains sequences necessary for the negative regulation of *osm-9* in response to developmental history.

## The TGF-β pathway negatively regulates *osm-9* in postdauer animals

To further illuminate the mechanisms regulating *osm-9* and ascr#3 avoidance behavior, we sought to identify factors that interact with the upstream regulatory sequences of *osm-9*. In addition to the potential DAF-3 SMAD binding site, further scrutiny of the *osm-9* cis-regulatory sequences yielded potential binding sites for the DAF-12 nuclear hormone receptor (NHR), which functions downstream of TGF-β and insulin signaling pathways in the regulation of dauer formation (*Figure 2A*). To investigate a potential role of the TGF-β and insulin/IGF dauer formation pathways in the regulation of *osm-9*, we examined *osm-9*p::*gfp* expression in CON and PD adults of strains carrying mutations in genes implicated in dauer formation. First, we found that mutations in *daf-3* SMAD and *daf-5* SNO/SKI significantly increased the *gfp* expression observed in PD ADL neurons compared to wild-type animals (*Figure 3A*). DAF-3 binds to DAF-5 (*da Graca et al., 2004*), and may act together as negative regulators at the *osm-9* locus. In contrast, the *daf-7* TGF-β mutant strain exhibits a partial but significant decrease in *osm-9*p::*gfp* expression in ADL neurons of both CON and PD animals compared to wild-type, but expression in PD animals is again down-regulated as compared to CON animals (*Figure 3A*). This result suggests that dauer formation, and not reduced TGF-β signaling alone, is required for the down-regulation of *osm-9* in PD ADL neurons. In addition, since DAF-7 TGF-β signaling antagonizes DAF-3/DAF-5 function, this result is also consistent with increased DAF-3/DAF-5 activity in a *daf-7* mutant promoting down-regulation of *osm-9* expression (*Vowels and Thomas, 1992*; *Thomas et al., 1993*; *Patterson et al., 1997*). Together these results suggest that DAF-3 and DAF-5 play a role in the developmental programming of *osm-9* in PD ADL neurons.

We next asked whether DAF-3 directly interacts with *osm-9* cis-regulatory sequences. The potential DAF-3 binding site in the PD motif (GTCTA; overlapping with Mutations 2 and 3, *Figure 2B*) differs from the previously identified binding site by a single base (GTCTG) (*Thatcher et al., 1999*). To test whether DAF-3 does indeed bind to the *osm-9* PD motif, we performed protein immunoprecipitation of DAF-3 from whole animals using a commercially available antibody (Novus Biologicals, Littleton, CO), followed by quantitative PCR (IP-qPCR) in CON and PD adult, dauer, and larval L2/L3 animals. Our results showed that DAF-3 was enriched at the potential DAF-3 binding site (DBS) in the *osm-9* promoter in PD adults compared to other tested stages (*Figure 3B*). We also examined DAF-3 enrichment at the *daf-8* and *daf-14* promoters as positive and negative controls, respectively, since their regulation by DAF-3 has been characterized previously (*Park et al., 2010*). Although these results do not address DAF-3 enrichment at the PD motif specifically in ADL neurons, nor the precise binding site of DAF-3 (see *Figure 3—figure supplement 1*), they are consistent with our model that DAF-3 negatively regulates *osm-9* in PD ADL neurons through a direct interaction at the PD motif.

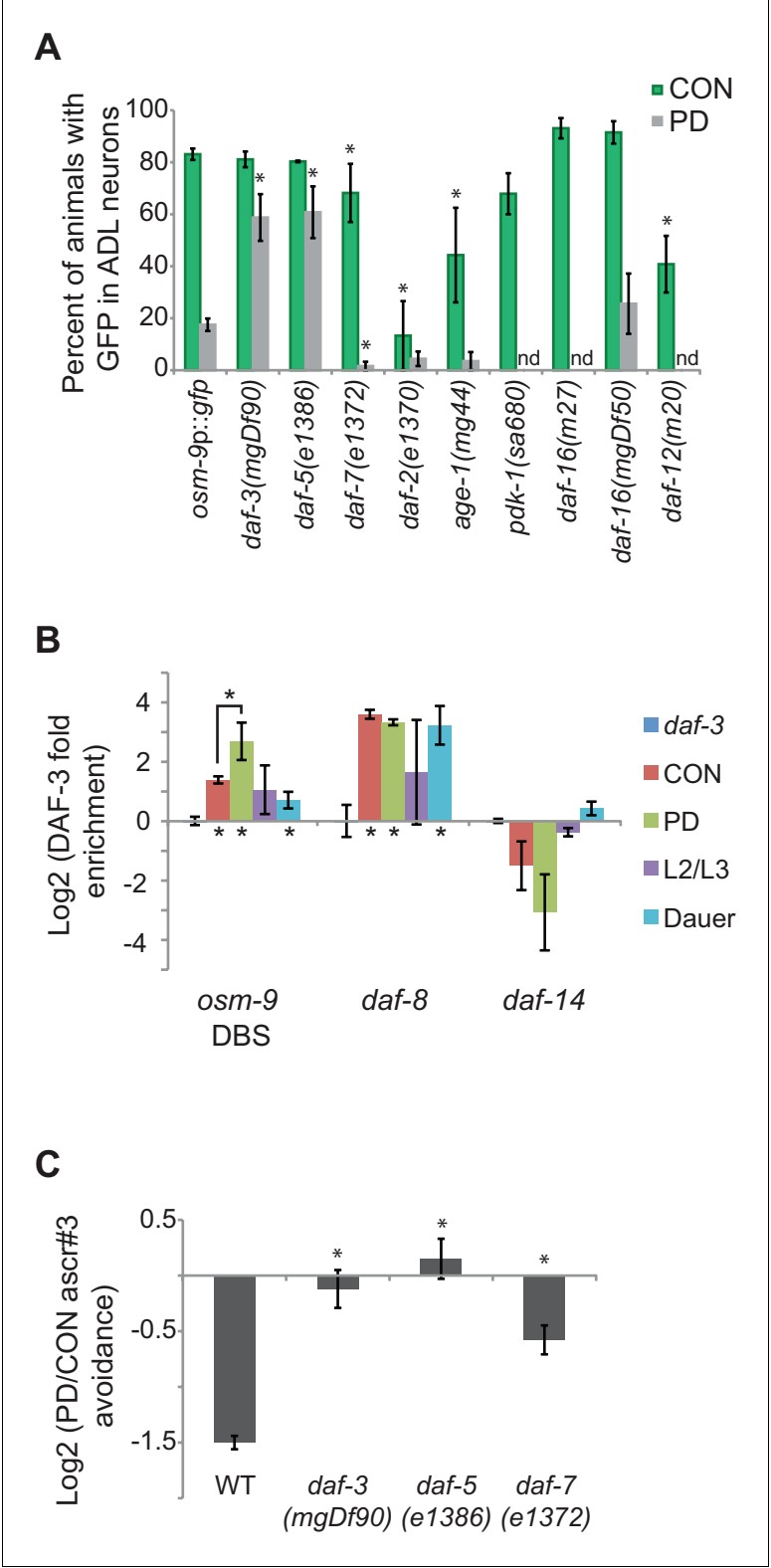

**Figure 3.** The TGF-β pathway regulates *osm-9* expression in response to developmental history. (**A**) Percent ADL neurons expressing *osm-9*p::*gfp* in CON and PD adults in wild-type and strains carrying mutations in TGF-β and insulin signaling pathways. GFP expression for Mutation 2 in *daf-3*, *daf-5*, and *daf-3; daf-5* mutants is found in *Figure 3—figure supplement 1*. N ≥ 3 trials; n ≥ 60 animals (*Figure 3—source data 1*). * indicates CON or PD significantly different from wild-type; One-way ANOVA with LSD posthoc correction, p<0.05. n.d., not determined. *Figure 3 continued on next page*

*Figure 3 continued*

(**B**) Log2 normalized enrichment of DAF-3 SMAD binding to the *osm-9* DAF-3 binding site (DBS) in *C. elegans* developmental stages. *daf-8* and *daf-14* are positive and negative controls, respectively. Bar graph represents IP-qPCR data, normalized to DAF-3 binding to actin *act-2* promoter (**Park et al., 2010**). N ≥ 2 biologically independent trials (**Figure 3—source data 2**). * indicates enriched over background (*daf-3* CON) or between CON and PD as indicated, p<0.05, Student's t-test. (**C**) Log2 PD/CON ratio of ascr#3 avoidance behavior in wild-type and strains carrying mutations in TGF-β pathway genes. Data for controls is found in **Figure 3—figure supplement 2**. * indicates significantly different from wild-type; One-way ANOVA with LSD posthoc correction, p<0.05 (**Figure 3—source data 3**). All error bars represent S.E.M.

The following source data and figure supplements are available for figure 3:

**Source data 1.** Spreadsheet containing percentages of animals expressing *osm-9*p::*gfp* in ADL neurons in strains carrying mutations in TGF-β and insulin signaling genes.

**Source data 2.** Spreadsheet containing DAF-3 enrichment values in wild-type CON, PD, larval L3, and dauer animals.

**Source data 3.** Spreadsheet containing PD/CON ascr#3 avoidance ratios for strains carrying mutations in TGF-β and insulin signaling genes.

**Figure supplement 1.** DAF-3 and DAF-5 may also indirectly regulate *osm-9* transcription.

**Figure supplement 1—source data 1.** Spreadsheet containing the percentage of animals expressing Mut2 version of *osm-9*p::*gfp* in *daf-3(mgDf90)* and *daf-5(e1386)* strains.

**Figure supplement 2.** Strains tested for ascr#3 avoidance exhibited expected behaviors for positive and negative controls.

**Figure supplement 2—source data 1.** Spreadsheet containing avoidance responses CON and PD adults to M13 and 1 M glycerol.

We next characterized the role of the TGF-β pathway in modulating ascr#3 avoidance behavior by testing the ascr#3 avoidance responses of CON and PD animals in strains carrying mutations in *daf-3* SMAD, *daf-7* TGF-β, and *daf-5* SNO/SKI genes. Based on their role in the regulation of *osm-9* transcription (**Figure 3A**), we expected to observe an elimination of the difference between the PD and CON ascr#3 avoidance behavior in *daf-3* and *daf-5* mutant strains and a reduced effect in *daf-7*. Indeed, we observed that mutations in *daf-3* SMAD and *daf-5* SNO/SKI genes eliminated the difference between CON and PD avoidance response to ascr#3, consistent with DAF-3 and DAF-5 acting as negative regulators of ascr#3 avoidance in PD animals (**Figure 3C**). In addition, a mutation in the *daf-7* TGF-β gene significantly disrupted the PD/CON ascr#3 avoidance levels compared to wild-type, through a decrease in CON adult ascr#3 avoidance (**Figure 3C** and **Figure 3—figure supplement 2A**). While most components of the TGF-β pathway exhibit broad expression patterns, DAF-7 TGF-β is expressed in ASI, ADE, and OLQ neurons, suggesting that a molecular signal from one of these sensory neurons may regulate *osm-9* expression in ADL (**Ren et al., 1996**; **Schackwitz et al., 1996**; **Meisel et al., 2014**).

Since TGF-β and insulin signaling pathways act in parallel to regulate dauer formation, we examined whether members of the insulin signaling pathway also contribute to the developmental programming of *osm-9*. We examined *osm-9*p::*gfp* expression in animals carrying mutations in the *daf-2* insulin/IGF receptor homolog, *age-1* PI3K homolog, *pdk-1* 3-phosphoinositide-dependent kinase 1 ortholog, and *daf-16* forkhead FOXO transcription factor (**Friedman and Johnson, 1988**; **Vowels and Thomas, 1992**; **Kenyon et al., 1993**; **Gottlieb and Ruvkun, 1994**; **Morris et al., 1996**; **Paradis et al., 1999**). We observed that mutations in *daf-2* and *age-1* significantly reduced GFP levels in ADL neurons of CON adults compared to wild-type, suggesting that insulin signaling is a positive regulator of *osm-9* expression (**Figure 3A**). However, the *pdk-1* mutant strain and two different mutations in *daf-16* FOXO, which functions downstream in insulin signaling, exhibited wild-type levels of GFP in ADL neurons (**Figure 3A**). Furthermore, we tested *osm-9*p::*gfp* expression in a *daf-12*

NHR mutant strain, which has a potential binding site (GGTGTGAC) located 362 bp upstream of the *osm-9* translational start site (*Figure 2A*) (*Ao and Gaudet, 2004*). Interestingly, the *daf-12* mutant also exhibited significantly decreased GFP in CON adults compared to wild-type (*Figure 3A*). We were unable to obtain postdauers from the *pdk-1*, *daf-16(m27)*, and *daf-12* strains to examine PD *osm-9p::gfp* expression. TGF-β and insulin signaling pathways converge onto DAF-12, which integrates environmental signals and regulates gene expression programs by binding to different ligands (*Mahanti et al., 2014*; *Dansey et al., 2015*; *Wang et al., 2015b*). Our results suggest that insulin signaling, as well as DAF-12 NHR, does not play a role in the developmental programming of *osm-9*, but instead contributes to the positive regulation of constitutive *osm-9* expression. Together, these results suggest a model where TGF-β and insulin signaling, as well as DAF-12 NHR, promote constitutive *osm-9* expression in ADL neurons during favorable growth conditions, resulting in ascr#3 avoidance in adult animals. In unfavorable conditions, reduced TGF-β signaling in conjunction with dauer formation results in the programmed down-regulation of *osm-9* in PD ADL neurons through the action of DAF-3 and DAF-5, resulting in the failure of adult animals to avoid ascr#3 (*Figure 3C*).

## *Mutator* proteins and nuclear RNAi silencing pathways play a role in the developmental programming of *osm-9*

Although our data thus far strongly indicate that the developmental programming of *osm-9* is established by upstream regulatory sequences, the inability to rescue the ascr#3 avoidance behavioral defect in PD adults of wild-type or *osm-9* mutants upon expression of *osm-9* coding sequences suggests that the coding sequence may also be a target of developmental programming (*Figure 1D*). In plants, fungi, and animals, short interfering RNAs (siRNAs) have been shown to act in *trans* to silence homologous sequences either by inhibiting transcription (transcriptional gene silencing, TGS) or through the destruction of mRNA (post-transcriptional gene silencing, PTGS) (reviewed in *Castel and Martienssen, 2013*). To examine whether endogenous RNAi pathways play a role in the developmental programming of *osm-9*, we examined *osm-9p::gfp* expression in a subset of strains carrying mutations in genes with known functions in RNAi pathways (*Figure 4A* and *Figure 4—figure supplement 1*). These strains included animals mutant for components of the *Mutator* focus, a protein complex that localizes adjacent to P-granules and is thought to contribute to siRNA amplification in the germline (*Zhang et al., 2011*; *Phillips et al., 2012*). However, whether *Mutator* proteins form a complex in somatic tissue and their functions in neurons remain unclear. We found that mutations in five of the seven proteins known to associate with the *Mutator* focus (*mut-2/rde-3*, *mut-14*, *mut-15*, *mut-16*, and *rrf-1* RdRP) resulted in a significant increase in GFP expression in PD ADL neurons compared to CON (*Figure 4A*). In previous work, we identified low abundance siRNAs mapping to the *osm-9* locus with a bias for exon sequences (*Hall et al., 2013*), indicative of siRNA biogenesis through the action of RdRPs such as RRF-1 (*Figure 4—figure supplement 2*) (*Pak et al., 2007*; *Sijen et al., 2007*). To confirm that endogenous RNAi pathways are contributing to the down-regulation of *osm-9* in PD ADL neurons, we again performed smFISH to measure *osm-9* mRNA abundance in ADL neurons of CON and PD adults carrying a mutation in the *mut-16* gene. For all three trials performed, we observed a significant increase in the mean and maximum fluorescence intensities of PD ADL neurons in *mut-16(mg461)* strain compared to wild-type (*Figures 1C* and *4B* and *Figure 4—figure supplement 3*). When the trials were combined, the *mut-16(mg461)* strain exhibited only a 7% change in mean fluorescence intensity of PD/CON ADL neurons compared to a 42% difference in wild-type (*Figure 4B*), indicating that functional MUT-16 is required for the full down-regulation of *osm-9* mRNA levels in PD ADL neurons. As a control, we also measured *gfp* mRNA abundance in the *mut-16* strain and observed no significant differences in mean and maximum intensity levels between CON and PD ADL neurons (*Figure 4—figure supplement 4*). Together, these results indicate a role for the *Mutator* proteins in the PD control of *osm-9* expression in ADL neurons.

   To further investigate the role of *Mutator* proteins in the regulation of *osm-9* expression in PD animals, we tested whether the examined mutants affect the ascr#3 avoidance behavior in CON and PD animals. Indeed, we found that mutations in genes associated with the *Mutator* complex, *mut-15* and *mut-16*, eliminated the difference in ascr#3 avoidance between CON and PD adults (*Figure 4C*). The behavioral responses of *mut-2*, *mut-14* and *rrf-1* mutant strains also showed a trend towards disruption in the PD/CON ratio of ascr#3 avoidance (*Figure 4C*). These results indicate that a subset of the *Mutator* proteins contributes to the plasticity of ascr#3 avoidance behavior.

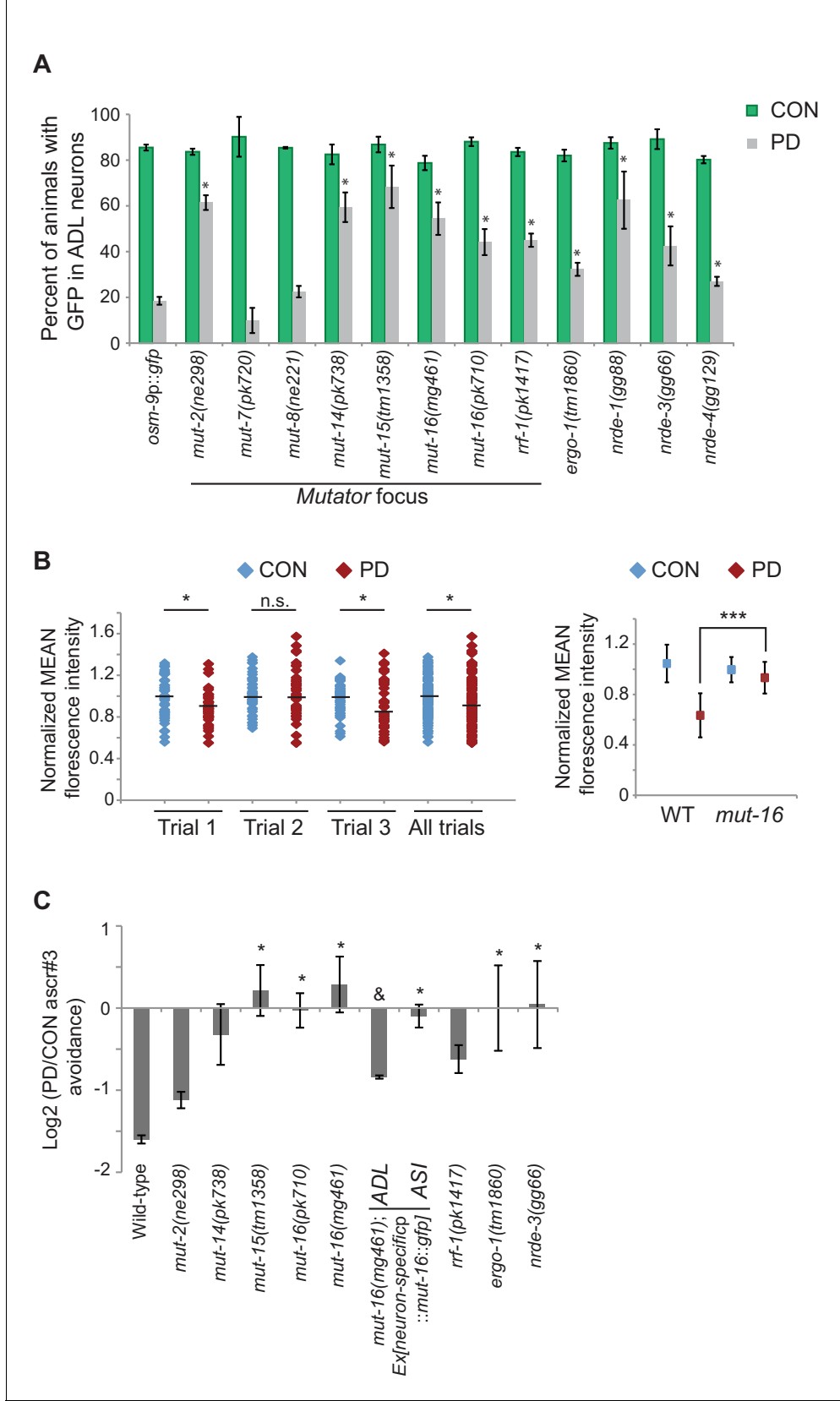

**Figure 4.** Endogenous RNAi pathways regulate the developmental programming of *osm-9* expression and ascr#3 avoidance behavior. (**A**) Percentage of ADL neurons expressing *osm-9*p::*gfp* in wild-type and strains carrying mutations in endo-RNAi genes. Data for additional mutant strains is found in *Figure 4 continued on next page*

*Figure 4 continued*

*Figure 4—figure supplement 1*. N $\geq$ 2 trials; n $\geq$ 40 animals (*Figure 4—source data 1*). * indicates mutant CON or PD significantly different from wild-type; One-way ANOVA with LSD posthoc correction, p<0.05. (**B**) Quantification of *osm-9* mRNA in ADL neurons of *mut-16(mg461)* CON and PD using smFISH. The graphs represent individual mean fluorescence measurements for N = 3 biologically independent trials; n $\geq$ 129 neurons (*Figure 4—source data 2*). Maximum florescent measurements are found in *Figure 4—figure supplement 3*. Control *gfp* mRNA measurements are found in *Figure 4—figure supplement 4*. Medians are indicated. Comparison of means with S.E.M. of all trials for CON and PD in wild-type and *mut-16* is shown. *p<0.05, ***p< $10^{-14}$, Student's t-test. (**C**) Log2 PD/CON ratio of ascr#3 avoidance behavior in wild-type and strains carrying mutations in endo-RNAi genes. Data for ADL and ASI-specific expression of *mut-16* rescue are averages of two extrachromosomal lines. N $\geq$ 3 trials; n $\geq$ 60 animals (*Figure 4—source data 3*). * indicates PD/CON ratio significantly different from wild-type; & indicates PD/CON ratio significantly different from *mut-16 (mg461)*; One-way ANOVA, LSD posthoc correction, p<0.05. All error bars represent S.E.M. Data for controls is found in *Figure 3—figure supplement 2*.

The following source data and figure supplements are available for figure 4:

**Source data 1.** Spreadsheet containing percentages of animals expressing *osm-9*p::*gfp* in ADL neurons in strains carrying mutations in genes with RNAi functions.

**Source data 2.** Spreadsheet containing raw and normalized mean florescence intensities of *osm-9* smFISH probes in ADL neurons in *mut-16(mg461)* strain.

**Source data 3.** Spreadsheet containing PD/CON ascr#3 avoidance ratios for strains carrying mutations in genes with RNAi functions.

**Figure supplement 1.** The ERGO-1/NRDE-3 endo-siRNA pathway contributes to the developmental programming of *osm-9* gene expression.

**Figure supplement 1—source data 1.** Spreadsheet containing percentage of animals expressing *osm-9*p::*gfp* in ADL neurons of additional strains carrying mutations in genes with RNAi and chromatin remodeling functions.

**Figure supplement 2.** *osm-9* siRNAs map predominantly to exons.

**Figure supplement 2—source data 1.** Spreadsheet containing siRNA sequences mapping to the *osm-9* locus for CON, PD, larval L3, and dauer small RNA libraries.

**Figure supplement 3.** *osm-9* mRNA levels quantified in *mut-16(mg461)* ADL neurons using smFISH.

**Figure supplement 3—source data 1.** Spreadsheet containing mean and maximum florescence intensities for *osm-9* smFISH probes in ADL neurons and worm backgrounds in *mut-16(mg461)* strain.

**Figure supplement 4.** *gfp* mRNA levels are unaltered between CON and PD ADL neurons in *mut-16(mg461)* strain.

**Figure supplement 4—source data 1.** Spreadsheet containing mean and maximum florescence intensities for *gfp* smFISH probes in ADL neurons and worm backgrounds in *mut-16(mg461)* strain.

Since dsRNAs are actively transported between cells and tissue types in *C. elegans* (*Feinberg and Hunter, 2003*; *Jose et al., 2009*; *Shih and Hunter, 2011*; *McEwan et al., 2012*; *Devanapally et al., 2015*), we asked whether the *Mutators* act cell-autonomously to regulate ascr#3 avoidance. To address this question, we expressed the *mut-16* cDNA under an ADL (*srh-220*p) or ASI (*gpa-4*p) specific promoter in the *mut-16(mg461)* strain and examined their ascr#3 avoidance phenotypes (*Figure 4C*). We chose to examine MUT-16 function in ASI neurons as well, since *daf-7* and other TFG-β pathway genes are expressed in ASI (*Ren et al., 1996*). We observed that cell-specific rescue of MUT-16 in ADL partially rescued the ascr#3 avoidance in CON and PD animals (*Figure 4C*). In contrast, MUT-16 expression in ASI failed to rescue the avoidance behavior (*Figure 4C*). This result suggests that MUT-16 functions in ADL neurons to mediate ascr#3 avoidance behavior due to developmental history.

Both the ERGO-1 and NRDE-3 endo-siRNA pathways require components of the *Mutator* focus and the ERI complex for the biogenesis of their associated siRNAs (*Duchaine et al., 2006*; *Pavelec et al., 2009*; *Gent et al., 2010*; *Zhang et al., 2011*; *Phillips et al., 2012*; *Thivierge et al., 2012*). ERGO-1 AGO associates with 26G-siRNAs to target mRNA transcripts in the cytoplasm and acts upstream of the 22G-siRNA-mediated NRDE nuclear RNAi silencing complex, which has been

shown to promote heterochromatin formation at targeted gene loci in somatic tissue (*Guang et al., 2008*; *Gent et al., 2010*; *Guang et al., 2010*; *Vasale et al., 2010*; *Burkhart et al., 2011a*; *Burkhart et al., 2011b*). Interestingly, we also observed a significant increase in PD GFP expression in ADL neurons in *ergo-1* AGO and nuclear RNAi silencing complex members, *nrde-3* AGO, *nrde-4*, and *nrde-1* mutants (*Figure 4A*). Moreover, the ERI complex also plays a role in the down-regulation of *osm-9* in PD ADL neurons (*Figure 4—figure supplement 1*). These results indicate that the *Mutator* proteins, along with the endo-siRNA ERGO-1/NRDE pathway are playing a role in the developmental programming of *osm-9*. Consistent with a role for the ERGO-1 and NRDE pathways in the regulation of *osm-9* expression in PD adults, we found that mutations in *ergo-1* and *nrde-3* also resulted in the elimination of ascr#3 avoidance differences between CON and PD adults (*Figure 4C*). Although we have not shown that *osm-9* is a direct target of endo-siRNA pathways, these results indicate that the *Mutator* proteins and ERGO-1/NRDE nuclear silencing pathways play a critical role in the developmental programming of *osm-9* and mediating the ascr#3 avoidance behavior due to developmental history.

## Chromatin remodeling pathways are required for maintenance of *osm-9* down-regulation in ADL neurons

Thus far, our results have suggested a model where *osm-9* is regulated via transcriptional gene silencing, which would promote alteration of the chromatin state at the *osm-9* locus. To ask whether functional chromatin remodeling pathways are necessary for the differential expression of *osm-9* we examined *osm-9*p::*gfp* expression in CON and PD animals of strains carrying mutations in genes involved in chromatin remodeling (*Figure 5A* and *Figure 4—figure supplement 1*). Although both SET-2 histone H3K4 methyltransferase and HDA-2 histone deacetylase act broadly to regulate histone modifications in the genome, their functions are antagonistic and correlate with gene activation or silencing, respectively (*Simonet et al., 2007*; *Hao et al., 2011*). We found that mutations in *set-2* and *hda-2* resulted in a significant increase in GFP expression in PD ADL neurons compared to wild-type (*Figure 5A*). Thus, our results suggest the possibility that SET-2 and HDA-2 are targeting additional gene loci that play a role in the down-regulation of *osm-9* in PD ADL neurons. We also examined the role of ZFP-1 in the regulation of *osm-9* in PD adults, since it has been shown to bind chromatin and is predicted to play a role in the regulation of actively expressed genes through interactions with both chromatin remodeling and RNAi pathways (*Grishok et al., 2008*; *Avgousti et al., 2013*). The known function of ZFP-1 is to negatively regulate RNA polymerase II elongation at loci of highly expressed genes (*Cecere et al., 2013*). Interestingly, we observed that a mutation in the *zfp-1* zinc finger protein gene eliminated the developmental programming of *osm-9*p::*gfp* expression in ADL neurons by significantly increasing GFP in PD adults (*Figure 5A*). Thus, ZFP-1 is an interesting candidate to play a role in the connection of the chromatin remodeling and the ERGO-1/NRDE-3 pathways in the developmental programming of *osm-9*.

Next, we tested the CON and PD adult ascr#3 avoidance response of strains carrying mutations in the genes *hda-2*, *set-2*, and *zfp-1*. We also examined the ascr#3 avoidance response of *Heterochromatin protein 1* homolog mutant, *hpl-2*, since HPL-2 has been shown to play a role in the NRDE-mediated silencing of the *odr-1* gene during olfactory adaptation (*Juang et al., 2013*). We observed that *hpl-2*, *set-2*, and *zfp-1* mutant strains exhibited an elimination or reversal of the wild-type PD/CON ascr#3 avoidance responses, and *hda-2* trended towards a reduction in the PD/CON ascr#3 avoidance behavior (*Figure 5B*). Together, these results suggest that functional chromatin remodeling pathways are required to maintain the down-regulation of *osm-9* in PD adults and mediate the ascr#3 behavior due to developmental programming.

Since chromatin remodeling proteins act broadly to regulate genes, we asked whether *osm-9* is a direct target of chromatin change. ZFP-1 is known to bind chromatin; thus, we tested whether *osm-9* is a direct target of ZFP-1. We created strains expressing the *zfp-1* gene tagged with *gfp* specifically in ADL neurons in a *zfp-1(ok554)* mutant. We found that expression of ZFP-1 in ADL partially rescued ascr#3 avoidance behavioral phenotype compared to the *zfp-1* mutant, indicating that the ZFP-1 transgene is functional (*Figure 5B*). Next, we performed chromatin immunoprecipitation of ZFP-1 from ADL neurons of CON and PD adults using α-GFP antibodies, followed by qPCR to test for enrichment of ZFP-1 at candidate gene loci. We found that ZFP-1 was significantly enriched at the *osm-9* DBS in PD adults, and enriched at *osm-9* coding sequences in both CON and PD ADL neurons (*Figure 5C*). We also examined ZFP-1 enrichment at the *act-3* and *egl-30* loci as positive

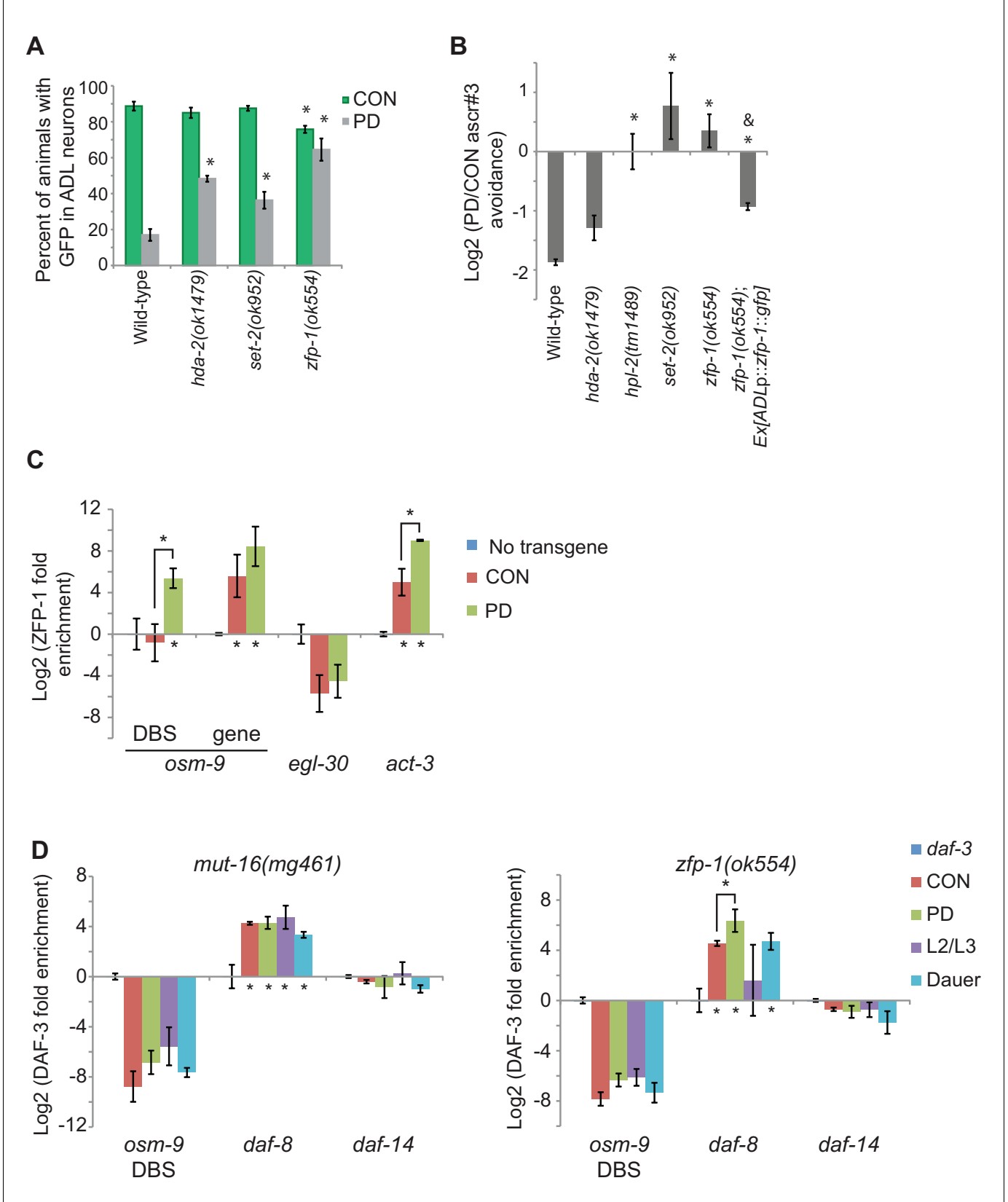

**Figure 5.** Chromatin remodeling pathways are required for developmental programming of *osm-9* and ascr#3 avoidance behavior. (A) Percentage of ADL neurons expressing *osm-9*p::*gfp* in wild-type and strains carrying mutations in genes with known chromatin remodeling functions. Data for *zfp-1*

*Figure 5 continued on next page*

*Figure 5 continued*

*(ok554)* represents average GFP expression of an *osm-9*p::*gfp* extrachromosomal array in two independent lines. N ≥ 3 trials; n ≥ 60 animals (*Figure 5—source data 1*). * indicates CON or PD significantly different from wild-type; One-way ANOVA with LSD posthoc correction, p<0.05. (B) Log2 PD/CON ratio of ascr#3 avoidance behavior in wild-type and chromatin remodeling mutant strains. Data for ascr#3 avoidance behavior of ADL-specific *zfp-1* rescue strains is an average of two extrachromosomal lines. N ≥ 3 trials; n ≥ 60 animals (*Figure 5—source data 2*). * indicates PD/CON ratio significantly different from wild-type; Student's t-test, p<0.05. & indicates PD/CON ratio significantly different from *zfp-1(ok554)*; Student's t-test, p<0.01. Data for controls is found in *Figure 3—figure supplement 2*. (C) Log2 enrichment of ZFP-1 at the *osm-9* DAF-3 binding site (DBS) and gene body in ADL neurons of CON and PD adults. Promoter regions for *egl-30* and *act-3* are positive controls (*Cecere et al., 2013*). Data was normalized to ZFP-1 binding to *gst-4* (negative control). N ≥ 2 biologically independent trials (*Figure 5—source data 3*). * indicates enrichment significantly greater than background or between PD and CON, Student's t-test, p<0.05. (D) Log2 normalized enrichment of DAF-3 SMAD at the *osm-9* DAF-3 binding site (DBS) in *mut-16(mg461)* and *zfp-1(ok554)* strains. *daf-8* and *daf-14* are positive and negative controls, respectively. Bar graph represents IP-qPCR data, normalized to DAF-3 binding to actin *act-2* promoter (*Park et al., 2010*), and adjusted to show enrichment of DAF-3 above background levels in *daf-3 (mgDf90)* strain. N = 3 biologically independent trials (*Figure 5—source data 4*). * indicates enrichment significantly greater than background or between PD and CON, Student's t-test, p<0.05. All error bars represent S.E.M.

The following source data is available for figure 5:

**Source data 1.** Spreadsheet containing percentages of animals expressing *osm-9*p::*gfp* in ADL neurons in strains carrying mutations in genes with chromatin remodeling functions.

**Source data 2.** Spreadsheet containing PD/CON ascr#3 avoidance ratios for strains carrying mutations in genes with chromatin remodeling functions.

**Source data 3.** Spreadsheet containing ZFP-1 enrichment values in ADL neurons of wild-type CON and PD adults.

**Source data 4.** DAF-3 immunoprecipitation data for the *osm-9, daf-8,* and *daf-14* loci in *mut-16(mg461)* and *zfp-1(ok554)* strains.

and negative controls, respectively. These loci have been shown to be ZFP-1 targets using whole worm samples (*Cecere et al., 2013*); however, ZFP-1 enrichment of these loci in ADL neurons was unknown. These results are consistent with ZFP-1 playing a role in the transcriptional gene silencing of *osm-9* in ADL neurons of PD animals.

Together, our results indicate a role for DAF-3 SMAD, ZFP-1, and endogenous RNAi pathways in the down-regulation of *osm-9* in ADL neurons due to developmental history (*Figure 6*). Mutations in individual genes with TGF-β, chromatin remodeling, and RNAi functions resulted in an increase in *osm-9*p::*gfp* expression in PD ADL neurons when compared to wild-type, suggesting that these pathways may be coordinating to silence *osm-9*. Since dauer formation is required for *osm-9* down-regulation, we hypothesized that DAF-3 SMAD binds to the *osm-9* promoter during dauer and recruits ZFP-1 and RNAi machinery to the *osm-9* locus to maintain the silencing to adulthood. To test our hypothesis, we again immunoprecipitated DAF-3 in *mut-16(mg461)* and *zfp-1(ok554)* strains and performed qPCR to examine for DAF-3 enrichment at the *osm-9* DBS. If DAF-3 binds to *osm-9* first and recruits ZFP-1 and RNAi machinery, we would expect to observe DAF-3 enrichment at the *osm-9* PD motif in mutant PD adults. However, our results showed that DAF-3 enrichment was significantly decreased in *mut-16* and *zfp-1* backgrounds at the *osm-9* DBS compared to wild-type (*Figure 5D*), while remained enriched at the *daf-8* promoter (positive control). These results indicate that DAF-3 enrichment at the *osm-9* DBS in PD animals is dependent upon functional ZFP-1 and MUT-16. Although we cannot yet address the interactions of these pathways specifically in ADL neurons, our results are consistent with a model that DAF-3 SMAD, ZFP-1, and endogenous RNAi pathways are acting cooperatively to down-regulate *osm-9* as a result of developmental history.

## Discussion

The results described here support a model in which the differential expression of the *osm-9* TRPV channel gene is dependent upon a transcriptional gene silencing mechanism regulated by the TGF-β, chromatin remodeling, and endogenous RNAi pathways (*Figure 6*). During continuous growth of *C. elegans* larvae in favorable conditions, DAF-7 TGF-β binds to the DAF-1 and DAF-4 receptors, resulting in a signaling cascade where DAF-3 SMAD is inhibited (*Thomas et al., 1993; Patterson et al., 1997*). Based on our results, we hypothesize that insulin signaling, DAF-12 NHR, and unidentified 'activators' act to promote transcription of *osm-9* in CON animals (*Figures 2,*

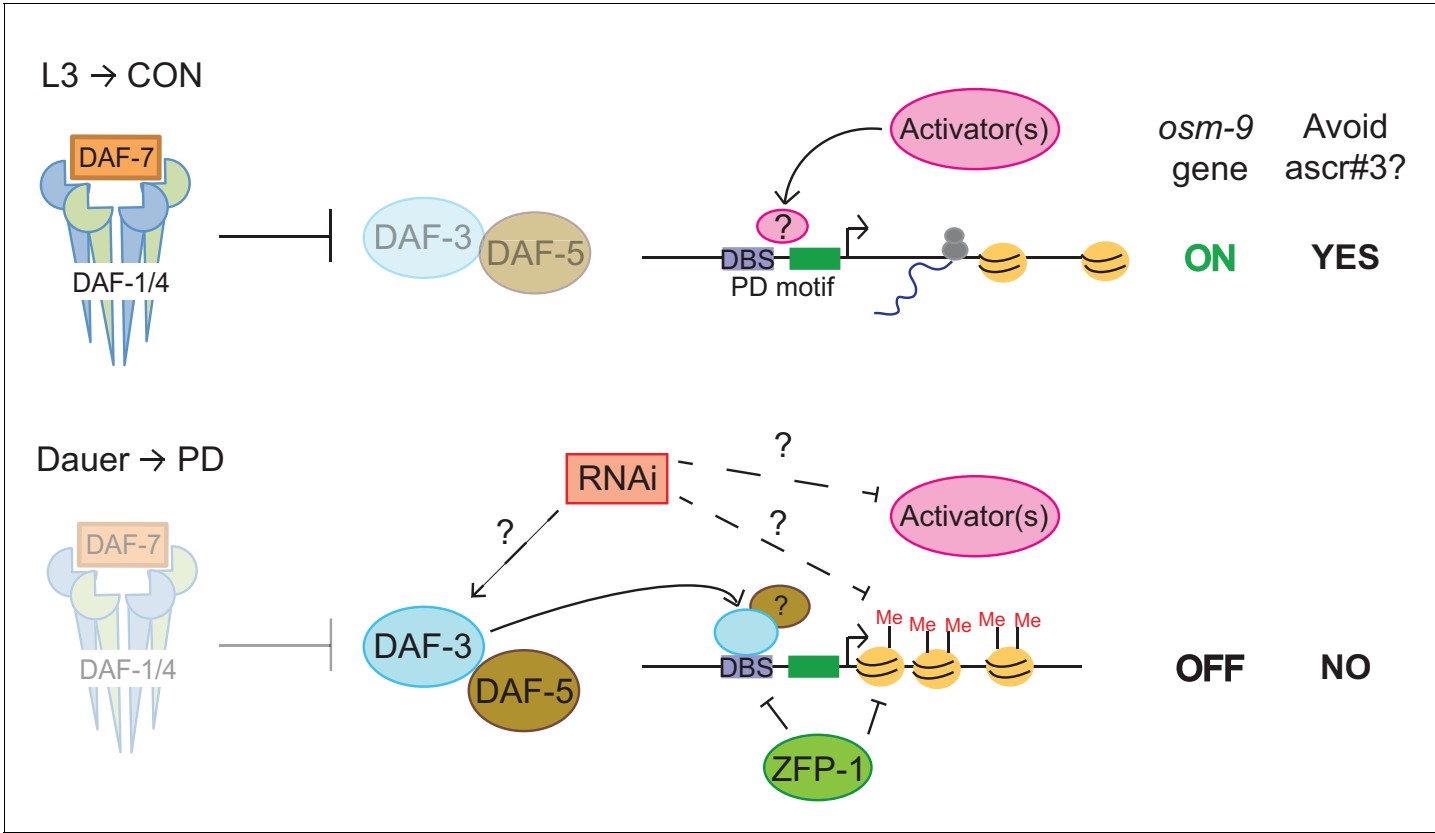

**Figure 6.** DAF-3, ZFP-1, and RNAi machinery act cooperatively to down-regulate *osm-9* in postdauer ADL neurons. Model for developmental programming of *osm-9* gene expression as a result of developmental history. See Discussion for details.

*3A*). Expression of OSM-9 TRPV channel in ADL neurons of CON adults results in the avoidance of ascr#3 pheromone (*Figure 1D*).

In stressful environments inducing dauer formation, *daf-7* TGF-β is not expressed, resulting in the de-repression of DAF-3. In this condition, we hypothesize that DAF-3, and possibly DAF-5, inhibit expression of *osm-9* in PD ADL neurons directly by binding to the PD motif (*Figures 2A*, *3A,B*). Our results showed that DAF-3 enrichment at the *osm-9* PD motif is dependent upon ZFP-1 (*Figure 5D*), which may be a direct interaction since both DAF-3 and ZFP-1 are enriched at the DBS in PD adults (*Figures 3B* and *5C*). ZFP-1 is also enriched in the gene coding sequence where it is predicted to promote chromatin remodeling and RNA polymerase II stalling (*Figure 5C*) (*Cecere et al., 2013*). Our results also show that endogenous RNAi pathways are required to maintain the transcriptionally silent state of *osm-9* and for DAF-3 enrichment at the PD motif, although we have not yet determined if this is due to direct and/or indirect targeting of the *osm-9* locus (*Figures 4A* and *6*). This maintained silent state of *osm-9* results in the failure to avoid ascr#3 pheromone in postdauer adults (*Figures 1D* and *6*). Thus, we postulate that phenotypic plasticity of olfactory behavior in wild-type adults is dependent upon developmental history, through modulation of *osm-9* gene expression by TGF-β, endogenous RNAi, and chromatin remodeling pathways.

## TGF-β and insulin signaling connect environmental conditions to regulation of *osm-9*

In our previous work, we identified 2127 genes that exhibited altered expression levels as a consequence of passage through the dauer stage (*Hall et al., 2010*). One question that arose from this observation was how the dauer developmental signal results in establishment of tissue-specific programmed changes in gene expression. DAF-3 SMAD and DAF-5 SNO/SKI are expressed throughout the worm and are required for dauer formation in unfavorable conditions (*Patterson et al., 1997*; *da Graca et al., 2004*). Here, we argue that DAF-3 and DAF-5 not only promote dauer formation,

but also establish altered expression levels of genes due to developmental programming. First, we showed that the differential expression of *osm-9* is dependent upon functional TGF-β signaling during dauer formation, and not just upon passage through the dauer stage (*Figure 3A*). In addition, we have shown that DAF-3 binds to the PD motif, and is required for the down-regulation of *osm-9* in PD ADL neurons (*Figures 3A,B*). Direct regulation by DAF-3 is an attractive model for how developmental history can result in established gene expression changes for target genes such as *osm-9*. Moreover, competitive binding to the PD motif between DAF-3 and other cell or tissue specific transcription factors can make the developmental programming specific to a cell type (such as ADL) without affecting expression in other cell types (such as AWA). A similar model of regulation by DAF-3 has been postulated for the *myo-2* gene in the pharynx (*Thatcher et al., 1999*). TGF-β signaling has also been shown to regulate neuronal receptor genes based on environmental cues via DAF-3-dependent and independent mechanisms in ASI, AWC, and interneurons (*Nolan et al., 2002*; *Lesch and Bargmann, 2010*; *McGehee et al., 2015*). Since 977 genes in the genome contain the PD motif and the DAF-3 binding site is enriched in chemoreceptor genes (*McCarroll et al., 2005*), we predict that DAF-3 might play a larger role in broadly establishing gene expression changes in response to environmental cues. We are currently testing our model by identifying additional proteins that bind to the PD motif in the *osm-9* gene and by examining expression patterns of other genes containing the PD motif.

## Endogenous RNAi pathways in neurons regulate phenotypic plasticity

Our results indicate that endogenous RNAi pathways regulate neuronal gene expression in response to developmental history. Temporal regulation of neuronal gene expression has been shown previously in AWC neurons, whereby DAF-3 establishes expression levels of chemoreceptor genes during early larval stages that are maintained in adults by the homeobox transcription factor ortholog, HMBX-1 (*Lesch and Bargmann, 2010*). We propose a similar model where DAF-3/DAF-5 contributes to the down-regulation of *osm-9* in animals that enter the dauer stage, which is dependent upon functional RNAi pathways (*Figure 5D*). At this time, we are unable to distinguish whether the requirement of MUT-16 for DAF-3 enrichment at the PD motif is indirect or direct. Our results are consistent with MUT-16 contributing to the positive regulation of *daf-3* expression, silencing of *osm-9* expression, or both (*Figures 4* and *5D*). We detected lowly abundant siRNAs homologous to *osm-9* in whole animals (*Hall et al., 2013*); however, the *daf-3* locus is targeted by significant numbers of siRNAs that require MUT-16 and CSR-1 AGO for biogenesis (*Claycomb et al., 2009*; *Zhang et al., 2011*), which is predicted to positively regulate gene expression by promoting a euchromatic chromatin state at target loci (*Youngman and Claycomb, 2014*). Furthermore, we find that components of the *Mutator* focus play a significant role in the differential regulation of *osm-9* in ADL neurons and modulation of ascr#3 avoidance behavior (*Figure 4*). To date, the localization and function of *Mutator* proteins in somatic tissue has been unclear, and some evidence suggests that *Mutator* component MUT-14 only functions within the germline (*Zhang et al., 2011*; *Phillips et al., 2014*). However, MUT-2, MUT-7, and the NRDE silencing complex have been shown to play a role in olfactory adaptation mediated by AWC neurons (*Juang et al., 2013*), and our results indicate that the *Mutator* proteins, MUT-2, MUT-14, MUT-15, and MUT-16, and the associated RdRP RRF-1, function to regulate *osm-9* expression in ADL (*Figure 4A*). In addition, we show that MUT-15, MUT-16, and RRF-1 contribute to modulation of ascr#3 avoidance behavior, and that MUT-16 is required in ADL neurons for the plasticity of ascr#3 avoidance due to developmental history (*Figure 4B*).

Moreover, we provide evidence that the ERGO-1/NRDE nuclear silencing pathways contribute to the maintained silencing of *osm-9* in postdauer animals (*Figure 4A* and *Figure 4—figure supplement 1*). Since the NRDE complex is associated with transcriptional gene silencing, our results suggest that *osm-9* regulation in PD ADL neurons by NRDE is more likely to be a direct interaction. We hypothesize that *Mutator*-amplified siRNAs direct the targeting of the NRDE nuclear silencing complex to the *osm-9* locus, which results in transcriptional gene silencing of *osm-9* in PD ADL neurons. Furthermore, we speculate that siRNAs generated to target the endogenous *osm-9* locus could also direct targeting of the NRDE complex to the ADL-specific *osm-9* rescue transgene, resulting in the transcriptional silencing of the *osm-9* coding sequence in the absence of the upstream regulatory sequences (*Figure 1D*). Although we have not shown directly that NRDE-3 targets *osm-9*, regulation by the NRDE complex is an attractive hypothesis, since we have evidence of both RNAi and transcriptional silencing mechanisms playing a role in regulating *osm-9* (*Figures 1A,B*, *2*, and *4A*).

Our results showing that the ERGO-1/NRDE pathway contributes to the down-regulation of *osm-9* in PD adults raises new questions of how RNAi pathways regulate endogenous genes due to developmental history. NRDE-3 must be bound by a siRNA to enter the nucleus and target nascent RNAs (*Guang et al., 2008*; *Guang et al., 2010*; *Burkhart et al., 2011a*; *Burkhart et al., 2011b*), which suggests a mechanism for selective siRNA amplification in animals with different developmental histories. This prediction is consistent with our previous findings that PD and CON animals have significantly different small RNA profiles (*Hall et al., 2013*). Further investigation of ERGO/NRDE pathways will be necessary to understand their interactions and roles in the maintenance of gene silencing due to developmental programming.

## Developmental programming of *osm-9* could contribute to environmental adaptation

Our results have led us to question why *osm-9* is down-regulated in PD animals. Dauer pheromone has been shown to have both primer and releaser effects on *C. elegans* larval development and adult behaviors, respectively (reviewed in *Ludewig and Schroeder, 2013*). We speculate that it is advantageous to down-regulate *osm-9* expression in animals that have entered the dauer stage. One possibility is that decreased *osm-9* expression during dauer results in a failure to respond to ascr#3, allowing worms to exit the dauer stage in areas of local high pheromone concentration. *C. elegans* animals that enter the dauer stage due to high population density and do not disperse would have their dauer exit suppressed by the local high concentrations of pheromone (*Golden and Riddle, 1982*; *1984*). Although ADL has not been implicated in dauer recovery to date, down-regulation of *osm-9* could allow dauer animals to become more sensitive to changes in their environment and allow faster dauer recovery when conditions improve (*Gruner et al., 2014*). Second, failure to avoid high concentrations of ascr#3 in postdauer hermaphrodites may serve as a mechanism for increased outcrossing in stressful environments. *C. elegans* males recover from long periods in the dauer stage at higher rates than hermaphrodites, increasing the opportunities for outcrossing in a postdauer population (*Morran et al., 2009*). We postulate that down-regulation of *osm-9* in postdauer hermaphrodites suppresses their dispersal to avoid high concentrations of pheromone, further increasing the opportunities for outcrossing. Thus, we speculate the differential regulation of *osm-9* via developmental programming mechanisms potentially serves to suppress both the primer and releaser effects of pheromone in order to promote survival and reproduction in stressful environmental conditions.

## Materials and methods

### *C. elegans* strains

All *C. elegans* strains were generated and maintained by using standard methods (*Brenner, 1974*; *Stiernagle, 2006*). Worm strains used in this study are described in *Supplementary file 1*. All strains were grown at 20°C or 15°C (for temperature sensitive strains) on NGM plates seeded with *E. coli* OP50. Developmentally synchronized control and postdauer adults were staged using egg plates as previously described (*Hall et al., 2010*; *Ow and Hall, 2015*). Controls were obtained by bleaching gravid adults to NGM plates without egg white. To induce worms by high crude pheromone concentrations, dauer formation assays were performed as previously described (*Neal et al., 2013*). Strains with dauer deficient phenotypes were exposed to a starvation protocol and grown at 25°C to produce dauers. To induce dauer formation by starvation, well-fed worms were plated onto seeded 100mm NGM plates, and were monitored daily for the depletion of food and the appearance of dauers. Dauer larvae were obtained by treating starved plates with 1% SDS for 24 hr.

### Imaging of *osm-9*p::*gfp* expression

*osm-9*p::*gfp* and *unc-122*p::*dsRed* were cloned into plasmids and microinjected as an extrachromosomal array at a concentration of 30 ng/μL into N2 and integrated into the genome using UV irradiation. The integrated *osm-9*p::*gfp* array (*pdrIs1*) was genetically crossed into mutant backgrounds for imaging. *osm-9*p::*gfp* was directly microinjected as an extrachromosomal array into *zfp-1(ok554)* strain since the integrated transgene and gene locus were closely linked. Animals were visualized on a Leica CTR550 microscope with a Hamamatsu C10600 camera. Adult animals were dye-filled with 2

mg/mL DiD for two hours prior to imaging in order to identify ADL neurons. A minimum of twenty animals was examined for CON and PD conditions for each strain over at least three trials on separate days. All animals were imaged as young adults, 24 hr after L4. Primer sequences for *osm-9*p::*gfp* cloning are located in *Supplementary file 2*.

## Mutagenesis

Mutagenesis of the *osm-9*p::*gfp* transgene was performed using the QuickChange XL kit (Agilent) following the specifications of the manufacturer. Deletions of the *osm-9* promoter were made by cloning individual promoter fragments into a plasmid containing *gfp*. All deletion and mutagenesis transgenes were microinjected as extrachromosomal arrays with co-injection marker *unc-122*p::*dsRed* into N2 at a concentration of 30 ng/µL and imaged as described above. Primer sequences are located in *Supplementary file 2*.

## Single molecule fluorescent in situ hybridization

smFISH experiments were performed essentially as described (*Ji and van Oudenaarden, 2012*). Briefly, CON and PD adults of the SH239 *otIs24 [sre-1p::gfp]* strain and SH265 *mut-16(mg461) I; otIs24* were fixed in 10% formaldehyde for 45 min followed by incubation in 70% ethanol for at least 48 hrs at 4˚C. Fixed worms were hybridized overnight at 30˚C with a set of 32 Stellaris *gfp* smFISH probes (labeled with Quasar 670 flourophores, Biosearch Technologies) (a kind gift from Oliver Hobert) and a set of 48 Stellaris *osm-9* smFISH probes (labeled with Quasar 570 flourophores, Biosearch Technologies). *osm-9* probe sequences are available upon request. Worms were mounted on slides using VectaShield H-1000 (Vector Laboratories).

Images for *osm-9* smFISH experiments were acquired on a Leica DM5500 B microscope coupled with a Leica CTR5500 electronic box mounted with a Hamamatsu Digital Camera C10600 ORCA R$^2$. The intensity of *osm-9* smFISH signals in the ADL neuron, as demarcated by *otIs24*, from maximally projected images of z-stacks (0.2 µM stacks) was quantified using the Leica LAS AF 3.1.0 software. Images of the *gfp* smFISH signals were taken on a Zeiss Axio Observer Z1 motorized microscope with Hamamatsu Orca Flash 4.0 LT camera followed by quantification on Image J. Exposure times were consistent within a trial, and images were not processed before analysis. Fluorescence quantification was performed blindly by two people. Normalization for individual measurements was performed by dividing each measurement by the median of the control samples for each trial. Since the signal to noise ratio was low (mean background > (mean – 2StDev)), we did not subtract background values from fluorescence measurements.

## Behavioral assays

The drop test acute avoidance response assay was performed using 100 nM ascr#3, M13 buffer (negative control), and 1M glycerol (positive control) as previously described (*Hilliard et al., 2002*; *Jang et al., 2012*). Animals were tested off food. For each strain, a minimum of three trials were performed on separate days, with a minimum of 30 animals assayed for each trial. The ascr#3 avoidance behavior was normalized by subtracting the proportion of animals responding to M13 buffer from the proportion of animals responding to ascr#3. For *Figure 1D*, the normalized avoidance response for each trial was averaged across trials for the graphs. For *Figures 3C*, *4C,* and *5B*, graphs represent averages of PD/CON ratios calculated for each trial. M13 and 1M glycerol avoidance data for each strain is found in *Figure 3—figure supplement 2*.

## Neuron-specific rescue strains

Overlap extension PCR (*Higuchi et al., 1988*) with Phusion DNA polymerase (NEB) was used to construct *sre-1*p::*mut-16 cDNA::gfp, gpa-4*p::*mut-16 cDNA::gfp, sre-1*p::*zfp-1 cDNA::gfp, gpa-4*p::*zfp-1 cDNA::gfp*, and *sre-1*p::*osm-9 cDNA::gfp* transgenes. To make *sre-1*p::*mut-16 cDNA::gfp*, 3 kb of the *sre-1* promoter, the *mut-16* cDNA (using a cDNA library prepared from total RNA of a mixed N2 population as the template), and the *gfp* gene (using Fire vector pPD95.75 as the template) were amplified separately. A final fused PCR was amplified using primers containing *Xma*I and *Sbf*I sites, digested with *Xma*I and *Sbf*I and cloned into pUC19. *gpa-4p::mut-16 cDNA::gfp* was constructed with the same method, except with 3 kb of the *gpa-4* promoter. ADL and ASI-specific *zfp-1* transgenes were cloned into pCR TOPO XL (Life Technologies) following the instructions of the

manufacturer. The fragment consisting of *sre-1p::osm-9 cDNA::gfp* was microinjected as a linear PCR at 30 ng/µl into N2 and *osm-9(ky10)*. Plasmids containing the *zfp-1* and *mut-16* neuron-specific rescues were microinjected (*Mello et al., 1991*) into N2, *mut-16(mg461)*, or *zfp-1(ok554)* animals at 50 ng/µl along with an *unc-122p::dsRed* co-injection marker (30 ng/µl) or genetically crossed into the desired mutant background. Primer sequences are located in *Supplementary file 2*.

## DAF-3 immunoprecipitation and qPCR

A packed pellet of worms (~100 µL) consisting of staged larval L2, dauer, CON, or PD animals were subjected to chromatin immunoprecipitation as described using 2% paraformaldehyde as the cross-linking reagent (*Hall et al., 2010*; *Hall et al., 2013*). Preparation of the lysate was done using a Sonic Dismembrator Model 100 (Fisher Scientific) and immunoprecipitation was performed using 5 µl of α-DAF-3 antibody (Novus Biologicals NB100-1924). DAF-3 IP was performed for three biologically independent samples for each condition. To determine if DAF-3 was bound to the *osm-9* PD motif, quantitative PCR was performed with 2–3 µl of the DAF-3 IP DNA using iTaq Universal SYBR Green Supermix (BioRad) following the recommendations of the manufacturer. Ct values were normalized using the *act-2* upstream regulatory sequences as previously described (*Park et al., 2010*). Primer sequences are located in *Supplementary file 2*.

## ZFP-1 immunoprecipitation and qPCR

A packed pellet of worms (~500 µL) consisting of control and postdauer young adult populations of SH198 *zfp-1(ok554) III; pdrEx22 [sre-1p::zfp-1 cDNA::gfp; unc-122p::dsRed]*, SH261 *mut-16(mg461) I; pdrEx22*, and SH262 *daf-3(mgDf90) X; pdrEx22* were subjected to chromatin immunoprecipitation as described previously using 2% paraformaldehyde as the crosslinking agent (*Hall et al., 2010*; *Hall et al., 2013*). Lysates were prepared using a Sonic Dismembrator Model 100 (Fisher Scientific). ZFP-1 immunoprecipitations were performed using 25 µl of GFP-nAb Magnetic beads (Allele Bio-tech) following the recommendations of the manufacturer. To determine ZFP-1 enrichment at gene loci, quantitative PCR was performed using 5 µL of the precipitated DNA using iTaq Universal SYBR Green Supermix (BioRad). Ct values for ZFP-1 enrichment were normalized to the *gst-4* gene, which is not a known target of ZFP-1. Promoter sequences for *egl-30* and *act-3* were positive controls for ZFP-1 enrichment since they have been identified as ZFP-1 targets (*Cecere et al., 2013*). Primer sequences are located in *Supplementary file 2*.

## Acknowledgements

We would like to thank Rebecca Butcher for the ascr#3 used for these experiments, Oliver Hobert for the generous gift of the *gfp* smFISH probes, Nikolaos Stefanakis and Scott Neal for assistance with the smFISH protocol, Austin Hager for chemotaxis data in *Figure 1—figure supplement 1B*, Torsten Woellert and George Langford for use of their microscope, Michael O'Donnell for critical reading of the manuscript, and CGC for strains (NIH Office of Research Infrastructure Programs, P40 OD010440). This work was supported by NIH NIGMS 5-F32-GM83593 and American Cancer Society 1092124-56171:IRG1105201 awards to SEH, Crown Wise Lynne Parker Scholar Award to MN, DGIST R&D Program of the Ministry of Science, ICT, and Future Planning (16-BD-06) to KK, and NSF IOS 0842452 to P.S.

## Additional information

### Funding

| Funder | Grant reference number | Author |
| --- | --- | --- |
| National Institutes of Health | NIGMS 5-F32-GM83593 | Sarah E Hall |
| National Science Foundation | IOS 0842452 | Piali Sengupta |
| National Research Foundation of Korea | 2012R1A1A2009385 | Kyuhyung Kim |

| DGIST R&D Program of the Ministry of Science, ICT and Future Planning | 15-BD-06 | Kyuhyung Kim |
| --- | --- | --- |
| American Cancer Society | 1092124-56171:IRG1105201 | Sarah E Hall |
| Syracuse University | Reneé Crown University Honors Program Lynn Parker Award | Mailyn A Nishiguchi |

The funders had no role in study design, data collection and interpretation, or the decision to submit the work for publication.

## Author contributions

JRS, MCO, MAN, Acquisition of data, Analysis and interpretation of data, Drafting or revising the article; KK, SEH, Conception and design, Acquisition of data, Analysis and interpretation of data, Drafting or revising the article; PS, Conception and design, Analysis and interpretation of data, Drafting or revising the article

## Author ORCIDs

Sarah E Hall, http://orcid.org/0000-0002-8536-4000

## Additional files

### Supplementary files

• Supplementary file 1. List of *C. elegans* strains used in this study.

• Supplementary file 2. Excel file containing primer sequences used in this study.

### Major datasets

The following previously published dataset was used:

| Author(s) | Year | Dataset title | Dataset URL | Database, license, and accessibility information |
| --- | --- | --- | --- | --- |
| Hall SE, Chirn GW, Lau NC, Sengupta P | 2013 | sRNA profiles of control adults, postdauer adults, larva L3, and dauer stages of C. elegans | http://www.ncbi.nlm.nih.gov/geo/query/acc.cgi?acc=GSE33313 | Publicly available at NCBI Gene Expression Omnibus (accession no: GSE33313) |

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
