## [Decision Letter]

Thank you for submitting your work entitled "Developmental programming modulates olfactory behavior in *C. elegans* via endogenous RNAi pathways" for consideration by *eLife*. Your article has been reviewed by three peer reviewers, and the evaluation has been overseen by a Reviewing Editor and a Senior Editor.

The reviewers have discussed the reviews with one another and the Reviewing editor has drafted this decision to help you prepare a revised submission.

Developmental imprinting is an important process by which early life experiences of an animal stably alters the organism's gene expression and thus physiology and perhaps psychology for life. The question of what molecular mechanisms trigger and maintain this changes is outstanding. In mammals, the circuits involved and their outputs are complex and poorly understood and thus invertebrate models with very simple circuits underlying imprinted behaviors are very attractive. This paper shows that an imprinted locus can be stably regulated by developmental experience by the action of the repressive co-SMAD DAF-3 binding to a specific region of the promoter. The authors used the *osm-9::gfp* as a convenient marker to mark this channel in the ADL neuron. Upon dauer formation via any of 3 mechanisms, the *osm-9* signal in the ADL neuron is eliminated. This ion channel is central to the olfactory sense in the worm, and consistent with this, the postdauer animals show reduced responses to a worm pheromone asc#3. The *osm-9::gfp* approach is clever, and the paper makes full use of worm genetics to narrow down the promoter region, and the mechanisms of transcriptional silencing in these animals. All three reviewers recognized the importance of the question and agree that the authors took the right approach to answer this question.

However, the reviewers did have some technical concerns that need to be addressed. We would like to invite the authors to respond to these questions in their resubmission.

First, the ChIP experiment lacks obvious controls. Second, the ChIP experiment lacks cellular resolution, which is important since the locus in question is expressed widely as is DAF-3. The authors also show that the repression of the reporter gene and the postdauer specific behavior requires the mutator locus and chromatin remodeling genes. The evidence for the involvement of these genes is purely genetic. The interpretation of these findings are not clear: could repression of OSM-9 be due to increased transgene silencing in the PD ADL neuron? Could the repression be due to increased DAF-3 down stream of mutator-induced repression of a DAF-3 repressor? They implicate direct interaction of the zinc finger protein ZFP-1 in repression of the promoter by showing it binds directly to the promoter but this does not prove the mutator locus is required for this factor's activity in the PD ADL.

Thus, to substantiate the claims the authors would need to perform the following experiments:

Experiment to link DAF-3 to siRNA by performing the Figure 3 ChIP in mutator/chromatin remodeling mutant backgrounds.

Experiment to provide cellular resolution to the repression: express *sre-1::daf-3*::3XFLAG and redo ChIP.

Experiment to link ZFP-1 Figure 5 to the mutator and *daf-3*. They should have separated control from PD and looked at ZFP-1 occupancy of the promoter/ gene in *daf-3* mutants, mutator mutant (*mut-16*) as well to ask if DAF-3 is required for ZFP-1 occupancy. This would provide at least an indirect link. Not perfect but approachable.

Experiment to link endosiRNA pathway to PD negative regulation of *osm-9* would be to do the smFISH in control vs. PD in *mut-16* background (at least one of the endosiRNA mutants).

ChIP experiments need negative controls: IP from the null background to examine non-specific IP; IP of an unaffected gene (*act-2* was not vetted anywhere in the paper).

Important but omitted: Better description of the *osm-9* endosiRNA profiles from control and PD… where do they map in the *osm-9* gene?

Other questions include:

Avoidance of the pheromone ascr#3 is tested, with the expected results. Can the authors provide control data to show that chemotaxis to unrelated molecules is normal? How specific is this effect-only to ascr#3 or other pheromones and nonpheromonal odorants?

Related to this, how pervasive is the effect of the postdauer experience on worm biology? The authors are using *osm-9::gfp*, allowing them to focus on a few neurons. But are other parts of the chemosensory circuit also affected by the developmental events? Some evidence for specificity would greatly enhance the conclusions.

In the last line of the subsection “The TGF-β pathway negatively regulates *osm-9* in postdauer animals” Figure 3 should be 3C.

Figure 5 authors should consider using an ampersand, as in Figure 4,

to indicate that the *zfp-1(ok554)::Ex(ADLp::zfp-1::gfp)* value differs from that of *zfp-1(ok554).*

In the first paragraph of the Discussion the authors may wish to reconsider their use of the term "transcriptional licensing".

[Editors' note: further revisions were requested prior to acceptance, as described below.]

Thank you for resubmitting your work entitled "Developmental programming modulates olfactory behavior in *C. elegans* via endogenous RNAi pathways" for further consideration at *eLife*. Your revised article has been favorably evaluated by a Senior editor, a Reviewing editor, and three reviewers. The manuscript has been improved but there are some remaining issues that need to be addressed before acceptance, as outlined below:

As you can see, reviewer 2 is still somewhat concerned by the responses provided in the last round of revision. Reviewer 3 has a specific request. Please address reviewer 3's request. Please also add the ADL-specific IPs results since you probably have had time to finish that. Once you have these results, we will consider the revised manuscript and provide a swift decision.

*Reviewer #1:*

The paper is does not pose any concerns. It does a good job of describing a very intriguing mechanism whereby a ubiquitous SMAD signal coupled with endogenous RNAi programs endow a neuron with plasticity. This is, as far as I know, only the second instance of an environmental cue acting via endogenous RNAi to program plasticity. Very nice.

*Reviewer #2:*

Sims et al. provide a revision that attempts to delve further into the mechanism by *osm-9* gene regulation is modulated after early life stress. Responses to the concerns that I raised in the initial review are acceptable. I continue to find the observations that developmental history reduces *osm-9* expression, and concomitant reduced sensitivity to pheromone, very interesting. Unfortunately, many of the experiments requested by one of the reviewers produced equivocal or incomplete results. In some cases, necessary controls were unavailable due to technical problems and time constraints for resubmission to *eLife*. Here is a case where the paper was actually weakened, rather than strengthened, through peer review. My sense is that there are a lot of interesting observations here, but there is a lack of conclusive data to support the mechanistic conclusions shown in Figure 6. The options would be to give the authors additional time to complete the missing experiments mentioned in the response to reviewers. Alternatively, the authors could back off from the strong conclusions, remove the incomplete data, and resubmit a paper more narrowly focused on the observations that seem robust, but removing the speculation not supported by the data.

*Reviewer #3:*

The comments I made in my earlier review have been satisfactorily addressed, except for one point that is not clear:

"The rescue strain expressing ADL-specific ZFP-1 is not significantly different in its behavior. In the manuscript, we describe the ascr#3 avoidance of the rescue strain as "partially restored".

If the rescue strain is not significantly different from the mutant, then it's not clear that it has been partially restored.

As for the comments made by the other reviewers, it seems that the manuscript has been improved. If it would be possible to improve the manuscript much more with a modest extension of the time line, then this would be OK with me.

---

## [Author Response]

[…] However, the reviewers did have some technical concerns that need to be addressed. We would like to invite the authors to respond to these questions in their resubmission.

First, the ChIP experiment lacks obvious controls. Second, the ChIP experiment lacks cellular resolution, which is important since the locus in question is expressed widely as is DAF-3. The authors also show that the repression of the reporter gene and the postdauer specific behavior requires the mutator locus and chromatin remodeling genes. The evidence for the involvement of these genes is purely genetic. The interpretation of these findings are not clear: could repression of OSM-9 be due to increased transgene silencing in the PD ADL neuron? Could the repression be due to increased DAF-3 down stream of mutator-induced repression of a DAF-3 repressor? They implicate direct interaction of the zinc finger protein ZFP-1 in repression of the promoter by showing it binds directly to the promoter but this does not prove the mutator locus is required for this factor's activity in the PD ADL.

Thus, to substantiate the claims the authors would need to perform the following experiments:

Experiment to link DAF-3 to siRNA by performing the Figure 3 ChIP in mutator/chromatin remodeling mutant backgrounds.

We agree with the reviewers that the cell-specificity of DAF-3 enrichment, and its dependence upon ZFP-1 and RNAi pathways, is important to the validity of our model of *osm-9* regulation. Since a strain carrying *sre-1*p::*daf-3::gfp* did not exist, we 1) cloned the ADL-specific *daf-3* transgene, 2) created stable, extrachromosomal transgene lines, 3) bulked the strains to obtain synchronous stage-specific populations, and 4) performed the IP-qPCR. In order to complete this experiment in a timely manner, we chose to inject wild-type, *mut-16*, and *zfp-1* strains directly with the transgene. We regret that we were unable to obtain a wild-type transgenic line on our first attempt, so that the data may be included in this resubmission. We now have this strain, and are performing the ADL-specific DAF-3 IPs currently. If the editors feel that this experiment is pivotal to our manuscript acceptance, we would happily include this data if granted an extension. We approximate ~ 3 weeks would be necessary to complete the ADL-specific IPs in wild-type animals. Instead, we have included the original DAF-3 enrichment data using an anti-DAF-3 antibody (Figure 3) with new positive and negative controls. Our results indicate that DAF-3 enrichment at the potential DAF-3 binding site is significantly higher than CON adults. This observation is consistent with our model that DAF-3 binds *osm-9* after dauer to facilitate silencing.

In Figure 6, we show ADL-specific DAF-3 enrichment in *mut-16* and *zfp-1* mutant backgrounds as requested. We were able to detect DAF-3 enrichment above background for the positive control in *zfp-1*, indicating that our experiment was successful. Interestingly, the results of this experiment indicate the novel finding that DAF-3 enrichment is dependent upon ZFP-1 activity. In addition, our data showed that both DAF-3 and ZFP-1 bind to the DAF-3 binding site region (Figure 3 and Figure 5), suggesting the possibility of a direct interaction between these two proteins. However, we did not see enrichment of DAF-3 in the *mut-16* background, even at the positive control locus (Figure 6). This observation could be due to experiment failure, or alternatively, could reflect the requirement of RNAi pathways to regulate *daf-3* expression as the reviewers suggested. Biogenesis of siRNAs mapping to the *daf-3* locus require functional MUT-16 and CSR-1 proteins (Zhang et al. 2011, Claycomb et al. 2009). Since CSR-1 AGO (bound by MUT-16 amplified siRNAs) promotes gene expression, we hypothesize that *daf-3* expression is decreased in a *mut-16* background. We have preliminary qRT-PCR data that support this hypothesis using mRNA from whole animals; however, additional experiments would be needed to examine MUT-16 dependent *daf-3* expression in ADL specifically. Our overall results cannot distinguish whether RNAi pathways affect *osm-9* regulation by promoting *daf-3* expression, silencing *osm-9* expression, or both. Our results examining the contribution of the NRDE complex to *osm-9* regulation, which promotes transcriptional gene silencing, is consistent with RNAi pathways acting to silence *osm-9* expression in PD ADL neurons and is not consistent with its potential regulation of *daf-3*. Thus, our current model is that RNAi pathways promote *daf-3* expression and silence *osm-9* expression through the action of different Argonaute proteins in PD ADL neurons. Our new model of *osm-9* regulation predicts that DAF-3 and ZFP-1 act cooperatively at the *osm-9* promoter to promote silencing.

*Experiment to provide cellular resolution to the repression: express sre-1::DAF-3::3XFLAG and redo ChIP.*

*Experiment to link ZFP-1 Figure 5 to the mutator and daf-3. They should have separated control from PD and looked at ZFP-1 occupancy of the promoter/ gene in daf-3 mutants, mutator mutant (mut-16) as well to ask if DAF-3 is required for ZFP-1 occupancy. This would provide at least an indirect link. Not perfect but approachable.*

Again, we agree with the reviewers that this is an important experiment to test our model of *osm-9* regulation. First, we examined ZFP-1 enrichment at the *osm-9* promoter and gene body in CON and PD wild-type animals. We found that ZFP-1 binds to the *osm-9* gene body in both populations, but shows significant enrichment at the DAF-3 binding site in PD compared to CON (Figure 5). This interesting result suggests the possibility that ZFP-1 and DAF-3 may be interacting directly in PD ADL neurons to promote silencing of *osm-9*. Our results showing that DAF-3 enrichment is dependent upon ZFP-1 supports this hypothesis (Figure 6). We also examined ZFP-1 enrichment at the *osm-9* locus in *mut-16* and *daf-3* strains, but we were unable to detect ZFP-1 enrichment above background for the positive controls (Figure 6—figure supplement 1). Since the *sre-1*p::*zfp-1::gfp* transgene was genetically crossed into these strains and worked well in the *zfp-1* background (Figure 5), we suspect that our unsuccessful experiment in the mutant strains was due to the presence of the endogenous ZFP-1 protein out-competing our tagged version. Ideally, we would need to generate *zfp-1; daf-3* and *zfp-1; mut-16* double mutants carrying the transgene to test this hypothesis, but we did not have the time. However, we were able to show that ZFP-1 differentially binds at the *osm-9* locus in ADL neurons due to developmental history, and contributes to the down-regulation of *osm-9* in PD ADL neurons.

*Experiment to link endosiRNA pathway to PD negative regulation of osm-9 would be to do the smFISH in control vs. PD in mut-16 background (at least one of the endosiRNA mutants).*

To further link siRNA pathways to the regulation of *osm-9* in PD ADL neurons, we repeated the smFISH experiment to examine *osm-9* mRNA levels in a *mut-16* strain. To perform this experiment, we genetically crossed the *sre-1*p::*gfp* transgene from wild-type (Figure 1) into the *mut-16(mg461)* strain. Our results show that *osm-9* mRNA levels in PD ADL neurons of *mut-16* are significantly increased compared to wild-type (Figure 4). These results nicely correlate with our *osm-9*p::*gfp* transgene results (Figure 4), and indicate that RNAi pathways are contributing to the down-regulation of *osm-9* (either directly or indirectly, see response #1) due to developmental history.

*ChIP experiments need negative controls: IP from the null background to examine non-specific IP; IP of an unaffected gene (act-2 was not vetted anywhere in the paper).*

We appreciate the reviewers’ suggestion to include additional controls for our ChIP experiments. To address their concerns, our resubmission includes ChIP-qPCR data for the *daf-3* null mutant background for DAF-3 IPs in wild-type (Figure 3) and no transgene controls for the ADL-specific ChIP experiments (Figure 5, Figure 6 and Figure 6—figure supplement 1). For each experiment, we normalized the ChIP-qPCR data to a negative control (not a known target) and adjusted the data to the null mutant/no transgene controls to better visualize enrichment. We indicate in the figures which IPs show significant enrichment above the null mutant/no transgene controls. We also included positive and negative controls for enrichment for each experiment. For DAF-3 IPs, we examined enrichment at the *daf-8* and *daf-14* as positive and negative controls, respectively (Figure 3 and Figure 6). We normalized this data to the negative control, *act-2*. These genes have been previously vetted as either targets or non-targets of DAF-3 (Park, Estevez, et al. 2010). For ZFP-1 IPs, we included *egl-30* and *act-3* genes as negative and positive controls, respectively (Figure 5 and Figure 6—figure supplement 1). Although ZFP-1 was shown to bind to *egl-30* in whole worms (Cecere et al. 2013), to our knowledge, *egl-30* does not show expression in ADL neurons (wormbase.org). *act-3* is a known ZFP-1 target and is expressed in ADL, so we examined this gene as a positive control (Cecere et al. 2013). The ZFP-1 data was normalized to enrichment at *gst-4* (not a known target) (Cecere et al. 2013).

*Important but omitted: Better description of the osm-9 endosiRNA profiles from control and PD… where do they map in the osm-9 gene?*

To address this concern, we reanalyzed small RNA sequencing data from CON, PD, L3, and dauer whole worms that were previously published (Hall et al. 2013). Unfortunately, we do not yet have small RNA sequencing data from ADL neurons specifically. In addition, the siRNAs mapping antisense to the *osm-9* gene were below the threshold cutoff for enrichment in our previous study; thus, we cannot make solid conclusions about the different numbers of siRNAs in CON versus PD ADL neurons that might be contributing to the differential regulation of osm-9. Despite these limitations, the numbers and mapping locations of the siRNAs homologous to the osm-9 locus in the different developmental stages are indicated in a new Figure 4—figure supplement 2. The results indicate that a majority of the siRNAs map to the 3’ exons of the *osm-9* gene, as expected for secondary siRNAs generated through the action of an RdRP. This analysis is consistent with our hypothesis that siRNAs generated in the endogenous ERGO-1/NRDE pathways are targeting *osm-9* for transcriptional gene silencing in PD ADL neurons.

*Other questions include:*

*Avoidance of the pheromone ascr#3 is tested, with the expected results. Can the authors provide control data to show that chemotaxis to unrelated molecules is normal? How specific is this effect-only to ascr#3 or other pheromones and nonpheromonal odorants?*

This is an excellent question from the reviewers that we believe we have addressed in our first submission. We have shown that attraction to the odorant diacetyl, which is an OSM-9 dependent and AWA mediated behavior, is unaffected by developmental history (Figure 1—figure supplement 1). Avoidance of 1 M glycerol, which is an OSM-9 dependent and ASH mediated behavior, is also unaffected in PD adults (Figure 3—figure supplement 2). We have not tested additional pheromone components, since CON adults to do not exhibit an avoidance response to the other major pheromones, such as C3 and C6 (Jang, et al. (2012) Neuron 75: 585-592). We believe that these experiments show that the reduced avoidance to ascr#3 is specific to the downregulation of *osm-9* in PD ADL neurons.

*Related to this, how pervasive is the effect of the postdauer experience on worm biology? The authors are using osm-9::gfp, allowing them to focus on a few neurons. But are other parts of the chemosensory circuit also affected by the developmental events? Some evidence for specificity would greatly enhance the conclusions.*

We believe that developmental history has profound effects on worm biology that we are just beginning to understand. We have shown previously that brood size and mean life span are significantly altered in PD adults compared to controls (Hall et al. 2010). In addition, we have found significant changes in mRNA levels of numerous genes expressed in neurons between CON and PD, and likely affect multiple behaviors which we hope to address in future work. However, to answer the reviewers’ question, ADL and ASH neurons are both electrically coupled to the RMG interneuron. We only observe significant changes in ADL mediated behavior between CON and PD, while ASH mediated behavior remains unaffected (see response #6). This result would suggest that the remainder of the circuit remains functional, and only ADL function is affected in this study.

*In the last line of the subsection “The TGF-β pathway negatively regulates osm-9 in postdauer animals” Figure 3 should be 3C.*

Thank you for pointing out this mistake. The correction has been made.

*Figure 5 authors should consider using an ampersand, as in Figure 4,to indicate that the zfp-1(ok554)::Ex(ADLp::zfp-1::gfp] value differs from that of zfp-1(ok554).*

Due to the variability in the ascr#3 avoidance response in the *zfp-1(ok554)* mutant, the rescue strain expressing ADL-specific ZFP-1 is not significantly different in its behavior. In the manuscript, we describe the ascr#3 avoidance of the rescue strain as “partially restored”.

In the first paragraph of the Discussion the authors may wish to reconsider their use of the term "transcriptional licensing".

The text has been changed.

[Editors' note: further revisions were requested prior to acceptance, as described below.]

The manuscript has been improved but there are some remaining issues that need to be addressed before acceptance, as outlined below:

As you can see, reviewer 2 is still somewhat concerned by the responses provided in the last round of revision. Reviewer 3 has a specific request. Please address reviewer 3's request. Please also add the ADL-specific IPs results since you probably have had time to finish that. Once you have these results, we will consider the revised manuscript and provide a swift decision.

Reviewer #2:

Sims et al. provide a revision that attempts to delve further into the mechanism by osm-9 gene regulation is modulated after early life stress. Responses to the concerns that I raised in the initial review are acceptable. I continue to find the observations that developmental history reduces osm-9 expression, and concomitant reduced sensitivity to pheromone, very interesting. Unfortunately, many of the experiments requested by one of the reviewers produced equivocal or incomplete results. In some cases, necessary controls were unavailable due to technical problems and time constraints for resubmission to eLife. Here is a case where the paper was actually weakened, rather than strengthened, through peer review. My sense is that there are a lot of interesting observations here, but there is a lack of conclusive data to support the mechanistic conclusions shown in Figure 6. The options would be to give the authors additional time to complete the missing experiments mentioned in the response to reviewers. Alternatively, the authors could back off from the strong conclusions, remove the incomplete data, and resubmit a paper more narrowly focused on the observations that seem robust, but removing the speculation not supported by the data.

Reviewer #3:

The comments I made in my earlier review have been satisfactorily addressed, except for one point that is not clear:

"The rescue strain expressing ADL-specific ZFP-1 is not significantly different in its behavior. In the manuscript, we describe the ascr#3 avoidance of the rescue strain as "partially restored".

If the rescue strain is not significantly different from the mutant, then it's not clear that it has been partially restored.

As for the comments made by the other reviewers, it seems that the manuscript has been improved. If it would be possible to improve the manuscript much more with a modest extension of the time line, then this would be OK with me.

Specific changes to the newly revised manuscript are described below.

1) One of the previous requests from the reviewers was to address the interaction of the TGF-β, chromatin remodeling, and RNAi pathways in the regulation of *osm-9* in postdauer ADL neurons. We previously attempted to answer this question by immunoprecipitating GFP-tagged DAF-3 and ZFP-1 from ADL neurons with little success due to technical and biological reasons. We have since performed DAF-3 immunoprecipitations in *mut-16* and *zfp-1* mutant strains using an α-DAF-3 antibody, and found that DAF-3 binding at the *osm-9* locus in postdauer animals is dependent upon functional ZFP-1 and RNAi pathways (Figure 5). To our knowledge, this would be the first report of an interaction between DAF-3 with ZFP-1 and RNAi pathways. Although this experiment does not address the interactions of these pathways in ADL specifically, our new results suggest that these pathways may be acting cooperatively at the *osm-9* locus to facilitate its down-regulation in postdauer animals. As reviewer #2 suggested, we have removed the negative results from the previous revision and simplified our model of *osm-9* regulation, emphasizing that we cannot yet say whether *osm-9* is a direct target of RNAi pathways (Figure 6).

2) As Reviewer #1 suggested, we have added the tRNA and non-coding RNA loci present in the *osm-9* intron to our siRNA map in Figure 4—figure supplement 2. Only 7 of the 143 siRNAs with homology to *osm-9* overlap with these loci. We have also included the siRNA nucleotide sequences mapping to the *osm-9* locus in Figure 4—figure supplement 2—Source data 1.

3) To address the concerns of Reviewer #3, we have performed an additional two trials to test the ascr#3 avoidance behavior of ADL-specific ZFP-1 rescue lines compared to *zfp-1* mutants. These trials were consistent with our previous data, and have improved the statistical power so that the ZFP-1 rescue lines now exhibit behavior that is significantly different from the *zfp-1* mutant line (Figure 5). Since the ascr#3 avoidance of the ZFP-1 rescue lines is also significantly different from wild-type, we have described this result as “expression of ZFP-1 in ADL partially rescued ascr#3 avoidance behavioral phenotype compared to the *zfp-1* mutant”.